ecology

deep refuge, marine heatwave,
benthic community composition, temperate,
foundation species

**Author for correspondence:**
Ana Giraldo-Ospina
e-mail: ana.giraldoospina@research.uwa.edu.au

# Depth moderates loss of marine foundation species after an extreme marine heatwave: could deep temperate reefs act as a refuge?

Ana Giraldo-Ospina[1,2], Gary A. Kendrick[1,2] and Renae K. Hovey[1,2]

[1]School of Biological Sciences, University of Western Australia, 35 Stirling Highway, Crawley, Western Australia 6009, Australia
[2]Oceans Institute, The University of Western Australia, 64 Fairway, Crawley, Western Australia 6009, Australia

AG-O, 0000-0003-4005-3548; GAK, 0000-0002-0276-6064; RKH, 0000-0003-0308-3609

Marine heatwaves (MHWs) have been documented around the world, causing widespread mortality of numerous benthic species on shallow reefs (less than 15 m depth). Deeper habitats are hypothesized to be a potential refuge from environmental extremes, though we have little understanding of the response of deeper benthic communities to MHWs. Here, we show how increasing depth moderates the response of seaweed- and coral-dominated benthic communities to an extreme MHW across a subtropical–temperate biogeographical transition zone. Benthic community composition and key habitat-building species were characterized across three depths (15, 25 and 40 m) before and several times after the 2011 Western Australian MHW to assess resistance during and recovery after the heatwave. We found high natural variability in benthic community composition along the biogeographic transition zone and across depths with a clear shift in the composition after the MHW in shallow (15 m) sites but a lot less in deeper communities (40 m). Most importantly, key habitat-building seaweeds such as *Ecklonia radiata* and *Syctothalia dorycarpa* which had catastrophic losses on shallow reefs, remained and were less affected in deeper communities. Evidently, deep reefs have the potential to act as a refuge during MHWs for the foundation species of shallow reefs in this region.

## 1. Introduction

In ecology, the term 'refugium' is used to describe regions that facilitate the temporal and spatial resilience of biological communities over evolutionary timescales or from past climate change [1,2]. Extreme climatic events (ECEs) can generate changes in species distributions, ecosystem structure and functioning [3–7] and are predicted to increase in magnitude and frequency because of climate change [8,9], and thus, taxa are being impacted in ecological timescales. To address the response of taxa to such ecological catastrophe, we have adopted the term 'refuge' to define spatial or temporal facilitation of environmental conditions or biotic interactions [2], that may enable the persistence of a species and/or communities in ecological timescales (years–decades) [2]. Climate change models have predicted shifts to the distribution of numerous species in terrestrial and marine ecosystems in response to climate change-related pressures [10–12]. However, species distributions are more affected by the local environment which in some cases may provide refuge and allow species to survive the climatic stress [13,14].

In the light of the increase in extreme climatic events (such as atmospheric and marine heatwaves (MHWs), droughts and wildfires) driven by climate change and their catastrophic consequences in marine environments [15–18], identifying refuges has become a research and conservation priority [2], because they have

the potential to prevent the extinction of local populations associated with extreme disturbance events. As such, deeper marine habitats have been identified as potential refuges for shallow-reef species [19–22]. Yet, the logistical difficulties of surveying deeper communities hinder our understanding of how these communities respond to extreme climatic events and whether their response varies from shallower reefs. Biogeographic transition zones are biodiversity hotspots owing to the overlap of taxa present at the edge of their biogeographical distribution (tropical and temperate taxa). These zones are particularly vulnerable to the disturbances of extreme climatic events as many species already live at the physiological limits of their distribution [23].

Oceanic MHWs are extreme climatic disturbances that are predicted to increase in frequency and intensity owing to climate change [8]. MHWs are defined as extended periods of anomalously high sea surface temperatures [24] which have already resulted in devastating effects on coastal marine communities characterized by widespread mortalities of invertebrates [3,25], seagrasses [26], coral, (associated with coral bleaching) [27], range contractions of habitat-forming species [28], and changes to community structure and ecosystem function [3,23]. MHWs have been documented across the globe: in the Mediterranean [29]; Australia [30]; northwestern Atlantic [31] and in the northeastern Pacific [32]. Understanding the response of marine ecosystems to MHWs is a key to predict their response to future climate change. Moreover, the ecosystem recovery from these impacts is variable and depends on processes such as population connectivity, fluctuations in fecundity, post-settlement success and altered species interactions.

In Western Australia during the summer of 2010/11, an extreme MHW superimposed over a general trend of ocean warming placed a global marine biodiversity hotspot at catastrophic risk [33]. This event was characterized by record high-temperature anomalies that extended across 12° latitude (Ningaloo Reef at 22° S to Cape Leeuwin at 34° S), up to 200 km offshore and down to depths of 50 m [33], with highest anomalies of +5°C around the central coast. The response of benthic marine communities to this extreme event is well documented for shallow habitats (less than 15 m depth). Kelp beds were lost across approximately 2300 km² causing a 100 km range contraction. Kelps and other macroalgae were replaced by less complex turf-forming algae and the recovery of kelp suppressed owing to the grazing pressure driven by an increase in tropical herbivorous fishes [23]. A staggering 1069 km² of the 4366 km² of seagrass meadows in Shark Bay were lost during and immediately after the 2011 MHW, resulting in significant ecosystem-wide changes [26,34], although recovery of 125 km² of meadows has occurred since 2014 [34]. At Ningaloo Reef, bleaching was observed in 79–92% of the coral cover [35] and at the Houtman Abrolhos Islands, the bleached coral was reported to be 6–42% varying across sites [36]. Despite all the evidence on the effect of MHWs on shallow marine ecosystems (less than 15 m), the response of deeper (greater than 15 m) benthic communities is often not documented and thus poorly understood.

As marine ecosystems continue to be degraded, identifying regions that can be a refuge for key species has become a priority for management and conservation [37]. Deeper marine reefs were first identified as refuges in the context of tropical coral reef ecosystems as research on mesophotic coral reefs increased [20,21,38]. Species living in deeper

habitats may benefit from a higher chance of survival from extreme environmental events owing to the buffering effect of depth [39,40]. Nonetheless, refuge habitats also need to share similar species with the habitat they are providing refuge for (in this case, shallow reefs). As a result, in the case of benthic species such as coral and algae, deep refuges are thought to be constrained to the upper regions of the mesophotic zone, typically shallower than 60 m [41]. However, most research on the ecology of deeper habitats and their potential role as refuges for shallow-water species has been focused in tropical coral reef ecosystems [41,42], while subtropical and temperate regions remain understudied.

In order to address the potential for deep habitats to act as refuges for temperate marine benthic communities, we aim to recognize if depth moderates the response of benthic communities before, during, and after an extreme MHW. We firstly characterize benthic community composition along a temperate to tropical biogeographic transition zone and across a depth gradient (15 m–40 m). Second, we describe how the community composition of deep benthic habitats was affected by the MHW (2010–2011) and how the response of deeper reefs compared to shallower reefs. Finally, we identify the key macroalgae species across the biogeographical transition zone and evaluate how their abundance changed with increasing depth and after the 2011 extreme MHW. We argue that deep benthic habitats in the biogeographic transition zone of Western Australia acted as refuges from the 2011 extreme MHW, as suggested by the reduced response of community composition and foundation species to this event. If depth helps foundation species moderate their response to future ECEs, these habitats may constitute refuges from future MHWs and from climatic change.

## 2. Material and methods

### (a) Benthic surveys and design

Benthic images were obtained from surveys conducted at permanent monitoring sites that were established by the Australian Integrated Marine Observing System (IMOS) initiative at the Houtman Abrolhos Islands, Rottnest Island and Jurien Bay [43] (figure 1). Surveys were conducted with an autonomous underwater vehicle (AUV) which records down facing georeferenced stereo image pairs, along with a suite of physical parameters including multibeam bathymetry, temperature, salinity, and chlorophyll a. Surveys were completed at two sites in each location, with an additional site at Abrolhos to account for a missing 25 m site, and at three depths: 15, 25 and 40 m (figure 1). Within each site, three replicate 'grids' were surveyed at the beginning of the monitoring programme in 2010. Each replicate 'grid' comprises a 625 m² area (25 × 25 m) of the seafloor surveyed by conducting parallel overlapping 25 m long transects across the seafloor. Grids at each location were located 50–200 m apart for spatial independence. Subsequently, surveys were repeated at every location every year until 2013, and the site Snapper Bank was added to Abrolhos Island. Only Abrolhos Islands were surveyed in 2014 and surveys were repeated at Abrolhos Islands and Rottnest Island in 2017 (electronic supplementary material, table S1). Repeated surveys aimed to assess the same three grids initially established, though on some occasions only one or two grids were surveyed owing to unfavourable weather conditions or equipment malfunction. More than 1000 stereo image pairs were captured by the AUV at each grid, but only 30 non-overlapping images (approx. 4 m²) per grid were randomly subsampled and processed as power analyses have indicated minimal improvement in detectable sizes for

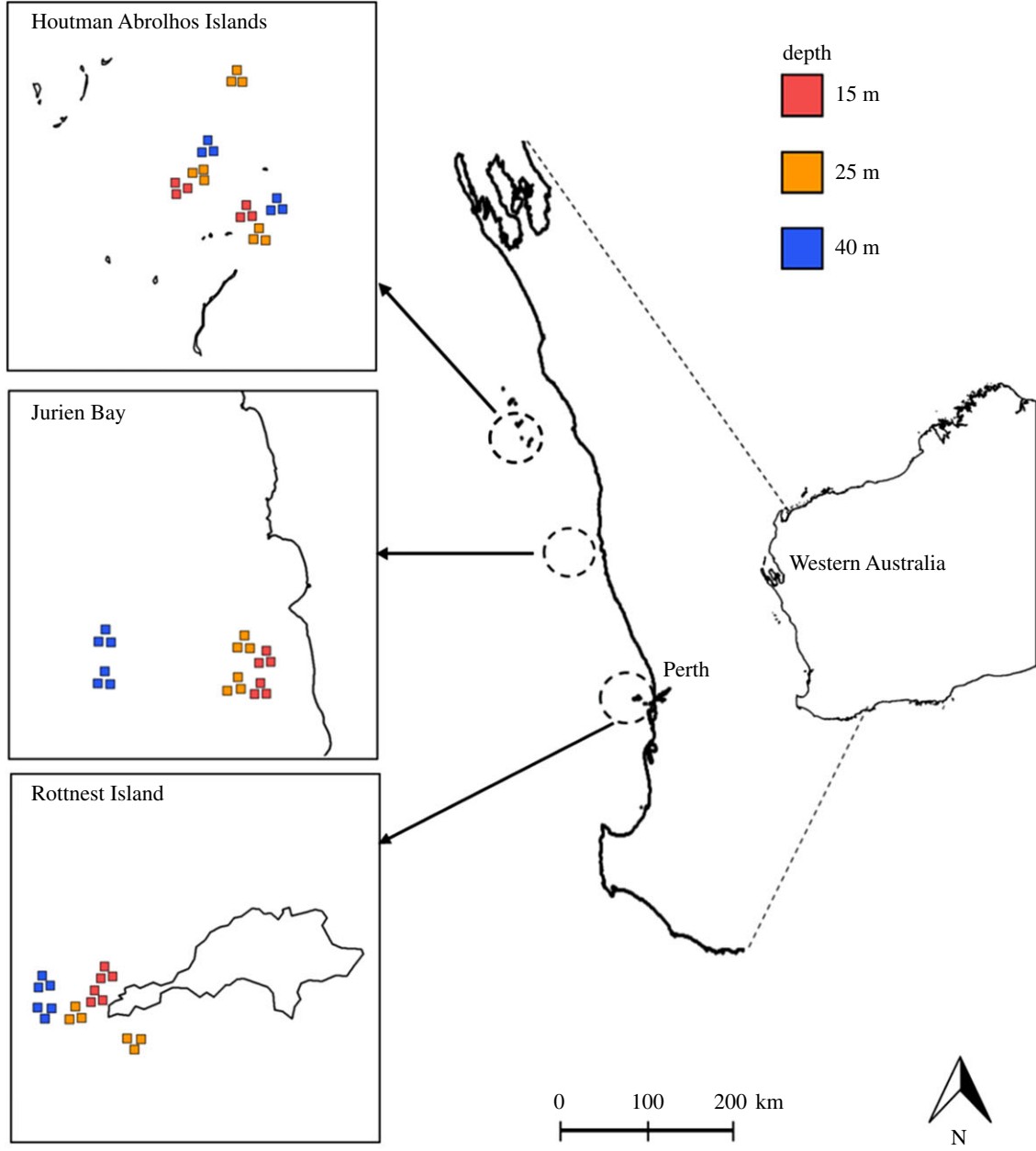

**Figure 1.** Locations of benthic surveys in Western Australia (Houtman Abrolhos Islands, Jurien Bay and Rottnest Island). The replicate grids are shown with respective depths, red: 15 m, yellow: 25 m and blue: 40 m. (Online version in colour.)

image replication above 30 [44]. Across all locations and times, this gave a total of 5970 images for analysis.

## (b) Study locations

The three locations surveyed in this study comprise the transition zone between the subtropical and temperate coast of Western Australia. The locations of Houtman Abrolhos Islands (28°43′S), Jurien Bay (30°29′S) and Rottnest Island (32°00′S) were chosen for long-term monitoring as they have been identified as key indicator regions because of their ecological importance as biogeographic transition zones and owing to their socio-economic importance in the region [43]. The Houtman Abrolhos Islands to the north form an archipelago 80 km from the mainland. The Abrolhos Islands are unique in their benthic assemblages, where patches of the kelp species *Ecklonia radiata* can be found coexisting with reef-building hard corals. Jurien Bay is situated between Abrolhos Island and Rottnest Island in a region characterized by inshore lagoons protected by offshore reefs and islands. The limestone reefs are dominated by *E. radiata*, other macroalgae, some corals and sponges. Rottnest Island in the south is located 19 km

from the mainland near the city of Perth. The island is surrounded by complex limestone reefs that are dominated by kelp species *E. radiata* and include numerous macroalgae species, seagrass meadows and coral.

## (c) Image classification

Each image was annotated by classifying the substrate, flora or fauna beneath 20 randomly and digitally overlaid points using CORAL POINT COUNT with EXCEL EXTENSIONS (CPCe) [45]. Each point was classified into functional/morphological groups with 104 categories in total, consistent with the Collaborative and Automated Tools for Analysis of Marine Imagery (CATAMI) classification scheme [46]. CATAMI provides a standardized vocabulary for image classification, enabling the compilation of regional, national and global datasets. Species of ecological importance were classified to species level and included *E. radiata*, *Scytothalia dorycarpa* and *Sargassum* sp. It is important to note that this method is poor at quantifying rare taxa or taxa smaller than 5 cm [47]. For each grid, subsampled images were pooled

and the analysis of community composition was conducted using the grids as spatially independent replicated units.

## (d) Data analysis

The multivariate community composition of the region was evaluated with a principal coordinate analysis (PCO) performed with Bray–Curtis similarity matrices based on square-root transformations of the data with a dummy variable (value of 1) used to optimize the year-to-year separation owing to a large number of zeroes in the data. The centroids represent means for each site per location, depth and year resulting from two or three grids. Community composition across depths was also examined at each location by constructing PCO plots. The centroids represent the averages for each depth per year derived from four to six grids. From this, we determined a trajectory of change in community composition in response to the 2011 MHW. In all PCOs, vectors over 0.7 correlation are illustrated to identify the benthic classes that characterize the assemblages.

Plots of change in per cent cover following the MHW (2010–2011) and compared to the latest survey (2010-last survey) were generated for *E. radiata*, turf, encrusting red algae and *S. dorycarpa*, so we could compare it to reported changes in inshore reefs [3] and to assess whether any recovery occurred. These plots were made by calculating the mean per cent cover of each species (or benthic class, like turf and encrusting red algae) at each grid, per location, depth and year, and then calculating the absolute change in per cent cover from 2010 to 2011 and from 2010 to the last survey (which varied with location, see the electronic supplementary material, table S1). Differences in per cent cover for *E. radiata*, turf, encrusting red algae and *S. dorycarpa* for each location and depth, were analysed by one-way analyses of variance (ANOVA) between 2010, 2011 and the year of the last survey, followed by a Tukey-test if differences were significant. When assumptions of normality and homogeneity of variance were violated, a Kruskal–Wallis test was used and Dunn's post-hoc test (electronic supplementary material, table S2).

Plots of mean per cent cover of principal benthic categories for each location, depth and year are presented to visualize their change in abundance through time and in response to the MHW. Certain benthic categories are only described for one or two locations, such as coral at Abrolhos Islands. At each location, differences in per cent cover of ecologically important benthic categories were tested between depths and across years with univariate PERMANOVAs, with depth (three levels) and year (six levels) as fixed factors. Data for each location consisted of the per cent cover of individual images, rather than the pooled grid averages used for the multivariate analyses. The tests used 9999 permutations of square-root transformed data and an Euclidean distance resemblance matrix.

## 3. Results

Three distinct community groups along the subtropical–temperate biogeographical transition zone of Western Australia were evident from the PCO: one for the shallow Abrolhos sites (15 m), one for Jurien and the deeper Abrolhos sites (25 and 40 m), and one for Rottnest Island (figure 2a,b).

The Abrolhos Islands were characterized by a mixed assemblage that varied markedly with depth (figure 2a,b). Shallow sites were dominated by reef-building corals of staghorn, tabulate, massive and foliose morphology and were colonized by the brown algae taxon *Sargassum* sp., while the deeper sites were either characterized by sparse *E. radiata*, *S. dorycarpa* or sand (figure 2a,b; electronic supplementary material, figure S1). Jurien Bay presented a community composition

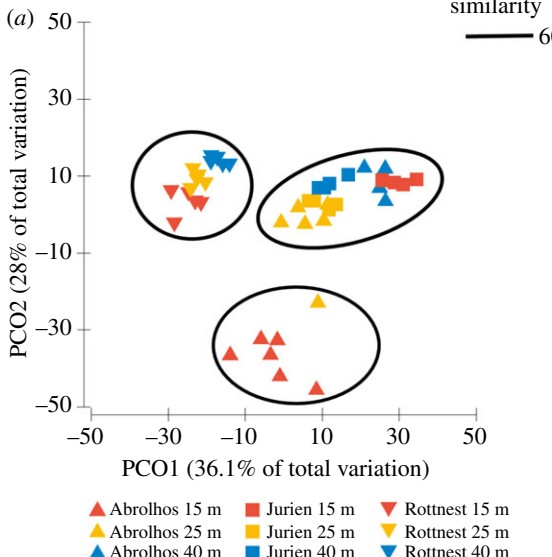

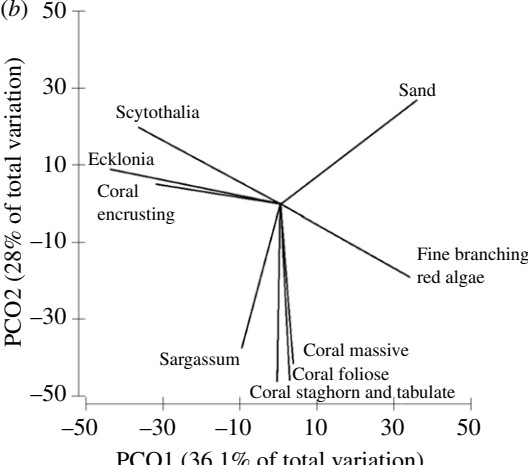

**Figure 2.** Principal coordinate analysis (PCO) of variation in benthic community structure at Abrolhos Islands, Jurien Bay and Rottnest Island based on a Bray–Curtis similarity matrix. The first two axes explain 64.1% of the variability in multivariate space. (a) Centroids represent average community composition at all locations for each year at each depth. Black rings indicate the centroids with 60% similarity. (b) Vectors indicating benthic categories with high correlations (Spearman correlation > 0.7) with axes. (Online version in colour.)

similar to the deeper sites of Abrolhos Islands (25 and 40 m) (figure 2a), characterized by a higher percentage of sandy substrate and fine branching red algae (figure 2b; electronic supplementary material, figures S1 and S4). Rottnest Island's community composition was dominated by large brown macroalgae at all depths, in particular, *E. radiata* and *S. dorycarpa* at the shallow sites, with encrusting red algae cover increasing with depth (figure 2a,b).

At Abrolhos Islands, there was a trend of greater change across years in community composition at 15 and 25 m, and less at 40 m (figure 3a). The only convergent community composition among years was shown between 2010 and 2017 at the 15 m site at Abrolhos. Following the MHW (2010 to 2011), the 15 m sites of Abrolhos Islands changed in community composition with an increase in bleached coral (approx. 4%) (electronic supplementary material, figure S2) and turf matrix (approx. 20%) (figure 4b; electronic supplementary material, figure S3), and a decrease in fine branching red algae

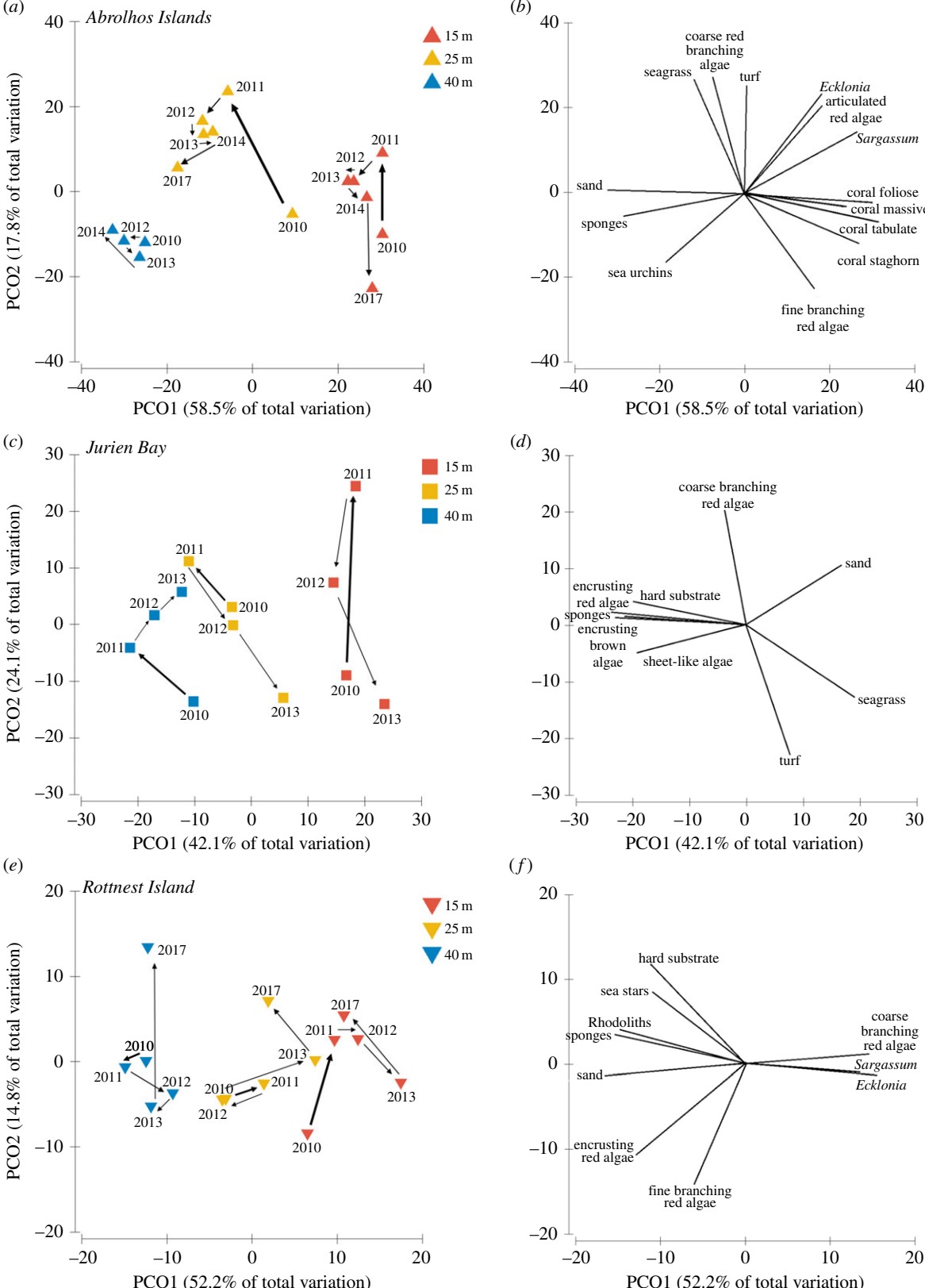

**Figure 3.** Principal coordinate analysis (PCO) of variation in benthic community structure at each location based on a Bray–Curtis similarity matrix. Centroids represent average community composition at each location ((a) Abrolhos Islands, (c) Jurien Bay, (e) Rottnest Island) for each year and depth and arrows indicate the trajectory. Thicker arrows show the change in average community composition from 2010 to 2011. Vectors ((b) Abrolhos Islands, (d) Jurien Bay, (f) Rottnest Island) indicate the benthic categories with high correlations with axes (Spearman correlation > 0.7). For Abrolhos, the first two axes explain 76.3% of the variability in multivariate space (a,b). For Jurien Bay, the first two axes explain 66.2% of the variability in multivariate space (c,d). For Rottnest Island, the first two axes explain 67% of the variability in multivariate space (e,f). (Online version in colour.)

(approx. 11%) (electronic supplementary material, figure S4) and foliose coral (approx. 5%) (electronic supplementary material, figure S5). Minimal change in encrusting red algae cover (approx. 2%) was seen in shallow sites (figure 4c;

electronic supplementary material, figure S6) and a decrease in *E. radiata* (approx. 3%) was observed (figure 4a; electronic supplementary material, figure S7). By 2017, the shallow (15 m) benthic community of Abrolhos Islands appeared to

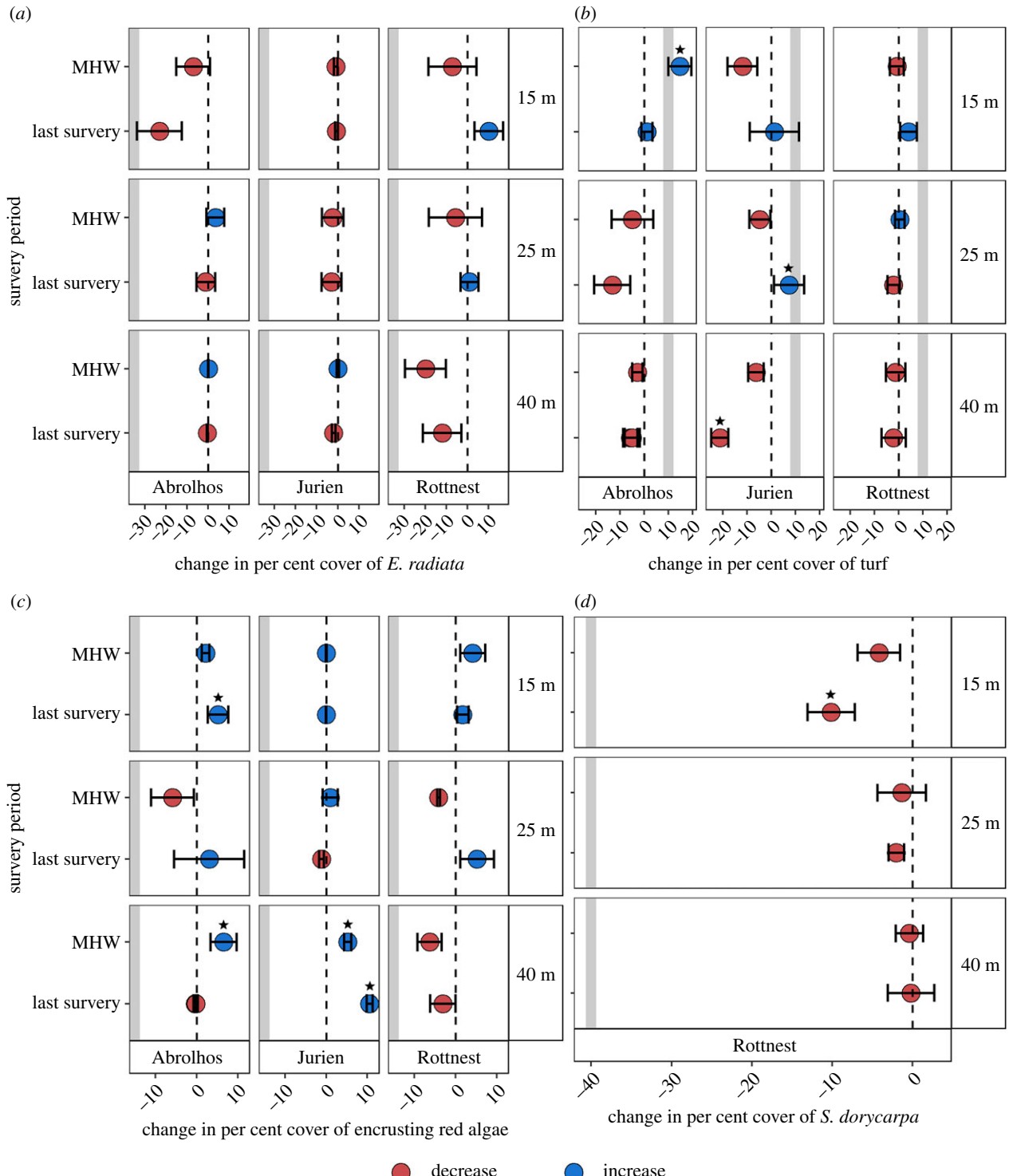

**Figure 4.** Absolute change in mean per cent cover (± s.e.) from 2010 to 2011 (heatwave) and from 2010 to latest survey for *E. radiata* (*a*), turf matrix (*b*), encrusting red algae (*c*) and *S. dorycarpa* (*d*) at each location (Houtman Abrolhos Islands, Jurien Bay, and Rottnest Island) and depth (15, 25 and 40 m). Colours describe an increase (blue) or decrease (red) in per cent cover in comparison to 2010. Significant changes in per cent cover are marked with a star. The grey box indicates the level of change reported for inshore reefs in response to the MHW [3]. The estimates of per cent cover are means of 2–6 grids (approx. 30 images per grid) within each location and depth per year. (Online version in colour.)

have returned to a state similar to pre-heatwave composition (figure 3*a*). The 25 m sites showed a response to the heatwave with an increase in *Sargassum* sp. (approx. 3%), *E. radiata* (approx. 10%) and seagrass (approx. 5%) (electronic supplementary material, figures S7–S9 respectively). By 2017, community composition had not returned to pre-heatwave conditions, with reduced turf cover (approx. 15%), increased encrusting red algae (approx. 3%), *Sargassum* sp. (approx. 3%), and seagrass (approx. 3%) (figure 3*a*). On the other

hand, there was minimal benthic community change at the Abrolhos 40 m sites (figure 3*a*) after the 2011 MHW and between all the years.

The Jurien Bay assemblage also changed across years, with the largest change occurring at the shallower sites (figure 3*c*) after the MHW. In contrast with the other locations, shallow sites at Jurien bay were more characterized by sand patches and seagrass. Seagrasses at this location showed large declines after the MHW (from approx. 8% to almost 0% cover)

and no signs of recovery at 15 and 25 m sites (electronic supplementary material, figure S9). The 25 and 40 m sites at Jurien also showed change after the MHW, mostly characterized by increases in encrusting algae (approx. 5% increase at both depths) (figure 4c). Communities at 25 m seemed to be affected by a separate event to the 2011 heatwave, since community composition was recovered to pre-heatwave conditions in 2012, and by 2013, it changed towards a more turf driven community (with an increase of approx. 20%).

The community composition at Rottnest Island also responded to the MHW at the shallow sites (15 m), with a reduced response in deeper sites (figure 3e). At the 15 m sites, there was a decrease in *S. dorycarpa* and *E. radiata* (approx. 5%, from 2010 to 2011 for both species) (figure 4a,d) and an increase in encrusting red algae (approx. 5%) (figure 4c). In contrast with the other two locations, community composition at shallow sites in Rottnest Island had not recovered to pre-heatwave conditions by 2017 (figure 3e). Cover of *S. dorycarpa* continued to decrease (approx. 10% decrease by 2013) and had not recovered by 2017 (figure 4d, electronic supplementary material, figure S10). Moreover, the analysis also identified changes in community composition in deeper habitats (25 and 40 m) that appeared to be a response to a process separate from the 2011 MHW, as they were observed from 2013 to 2017 (figure 3e). These changes were driven by an increase of approximately 5% encrusting red algae in the 25 m sites and an increase of approximately 4% in the cover of sponges and *S. dorycarpa* at 40 m sites (figure 3e,f; electronic supplementary material, figures S7, S10 and S11).

Despite changes in per cent cover of macroalgae following the MHW, these were not at the scale of the changes reported for inshore reefs (figure 4) [3]. Decreases of approximately 30% were reported in the cover of *E. radiata* owing to the MHW at shallow inshore reefs (figure 4a). We found the largest decrease in *E. radiata* cover at the deep sites of Rottnest Island to be of approximately 18% following the 2011 MHW and at no location or depth were these changes found to be significant (figure 4a). Turf cover increased by around 10% at shallow, inshore reefs after the 2011 heatwave [3], but we only found a comparable increase at the shallow sites of Abrolhos with a significant increase in turf cover of approximately 15% and at the 25 m sites of Jurien with a significant increase of approximately 8% (figure 4b). Other sites and depths did not show a large increase in turf cover after the 2011 MHW and a large significant decrease in turf cover (approx. 20%) was found in the deep sites of Jurien from 2010 to the last survey in 2017 (figure 4b). The largest decrease in encrusting red algae we observed at the 25 m sites at Abrolhos Islands, with a reduction of approximately 5%, but this was not significant, while in shallow inshore sites the reductions were of approximately 15% (figure 4c). Significant increases in encrusting red algae cover were observed in the 15 m and 40 m sites of Abrolhos (approx. 5% at both depths), and at the 40 m sites of Jurien with an increase of approximately 5% after the heatwave and a total of 10% by the time of the last survey in 2013 (figure 4c). *Scytothalia dorycarpa* at Rottnest Island showed the largest reduction at shallow sites (approx. 5%), and in the last survey, it presented a significant decrease (approx. 10%) compared to pre-heatwave levels, yet these reductions are small compared to the approximately 40% per cent cover decrease at inshore reefs (figure 4d).

At all locations, the variation in per cent cover of each benthic category was significantly different by year, depth and their interaction as indicated in multi-factorial univariate

PERMANOVA tests (electronic supplementary material, table S3). Exceptions were sponges at Jurien Bay, which did not exhibit an effect of year or its interaction with depth, and sand at Rottnest Island which showed no effect with the interaction of year and depth.

# 4. Discussion

MHWs are expected to increase in magnitude and frequency under climate change predictions, posing a threat to the persistence of numerous marine species [48]. Here, we showed evidence that there are potential refuges in deeper offshore reefs for shallow near shore foundation species in temperate regions, such as *E. radiata* and *S. dorycarpa*. These depth refuges add a dimension that has not been considered by many studies of widespread mortality on near shore shallow reefs [15,23]. The catastrophic loss of canopy-forming macroalgae, *E. radiata* and *S. dorycarpa* documented in shallow waters (less than 15 m) [15,23] was not shown from deeper offshore reefs between 25 and 40 m off Western Australia (figure 4a,d), supporting our hypothesis that deep water habitats exhibit a buffering effect from extreme climatic events, that allows the persistence of kelp-dominated communities.

Deep reefs in the mesophotic zone have been proposed to offer refuge from environmental disturbances [49,50], as suggested for tropical mesophotic coral ecosystems [21,38,51]. Here, we analysed benthic community composition across a subtropical–temperate biogeographic transition zone and found a reduced response to an extreme MHW in deeper reefs (25–40 m), despite a natural shift in community composition from mixed assemblages of tropical corals and kelps in the north (Abrolhos Islands) to a typical temperate community dominated by kelps in the south (Rottnest Island) [52,53]. Moreover, key habitat-forming taxa like *E. radiata*, and fine branching red algae, were found along the transition zone across all depths. *Scytothalia dorycarpa* was only found in Rottnest Island but showed small decreases in deep sites compared to in shallow ones. Although we observed decreases in *E. radiata* and *S. dorycarpa* which had not returned to pre-heatwave status by 2017, these reductions were minimal compared to the decreases reported for inshore communities following the 2011 MHW [3]. Foundation species persisting in deep reefs could provide a source of propagules for their shallow counterparts to facilitate the recovery of shallow disturbed populations [20], provided they are reproductive and have oceanographic connectivity. Because MHWs are predicted to become more frequent and intense in the future [8,9], deep reef communities may be a key driver of shallow-reef resilience, inasmuch as the frequency and magnitude of future MHWs allows for the recovery of shallow communities.

In this study, we found that benthic species in shallow offshore sites (15 m) were less affected by the MHW than what was reported in other studies [3,15,23]. For example, populations of *E. radiata* and *S. dorycarpa* suffered catastrophic losses in shallow, near shore reef ecosystems and resulted in a range contraction of approximately 100 km at the warmest edge of their distribution [23,28]. The loss of these key habitat-forming species further resulted in ecosystem reconfiguration driven by an increase in less structurally complex turf-forming seaweeds [23]. However, we did not see this regime shift in *E. radiata*-dominated communities at any depth or location in this study. These results have implications for the

spatial scope of benthic surveys and post-disturbance population or community assessments which take into consideration only the shallowest areas of the reef communities and consequently are focused only on the most susceptible area of the species distribution [3,23,28]. Additionally, the models of seaweed distribution along temperate Australia have shown that under ocean warming predictions there will be a significant poleward shift in distribution, with *E. radiata* being restricted to the south coast [54]. However, we have shown that habitat-building species living in deeper reefs have the potential to persist, and consequently, the range contractions suggested from modelling may have overestimated the total impact of climate change disturbances by not considering the differential response of deeper communities.

While temperature anomalies associated with the 2010–2011 MHW have been identified down to approximately 50–60 m of depth [33,55], we did not detect signs of catastrophic alteration in community composition as documented in shallower habitats (less than 15 m), as far south as Rottnest Island. We gathered sporadic, *in situ* temperature recordings near our study sites over a 20 year period, which also showed temperature anomalies at 40 m depths during the 2011 MHW (electronic supplementary material, figure S14); however, these data lacked enough replication over time to be used for further analyses. Benthic populations living in deeper reefs may be acclimated to frequent thermal variation owing to the effect of the Leeuwin Current which transports warm water from the tropics along the continental shelf of Western Australia [56] and consequently may have greater influence in deep offshore habitats than in shallow and inshore ones [57]. This high variability in water temperature may give them the capacity to withstand MHWs [58–60]. An additional coping mechanism for deep macroalgae may be enhanced photosynthetic efficiency owing to acclimation to lower light conditions, as opposed to their shallow counterparts which were exposed to high temperatures and higher light. This interaction between light and temperature has been shown in studies of the kelp *Laminaria saccharina* where adult sporophytes acclimated to high temperature and/or low light required less light to achieve positive net photosynthesis than sporophytes acclimated to low temperature or high light [61]. Furthermore, deeper communities, which are often found offshore, may be uncoupled from other co-occurring stressors that affect shallower, coastal ones. The interaction of multiple stressors in shallow coastal ecosystems has been shown to elicit extreme ecological responses via catastrophic loss of species [62] because the effect is synergistic, where the combined response is greater than the sum of individual stressors [63]. Our study did not find evidence for deep refuges in corals; however, this was because the deeper sites surveyed at Abrolhos Islands did not have substantial (greater than 1%) coral cover (electronic supplementary material, figure S2), so no buffering effect of depth could be inferred.

The temperate–subtropical biogeographic transition zone of Western Australia provides a model for understanding the effects of climate change on species distribution driven by an increase in sea temperature. Over geological time scales, the Leeuwin current has undergone periods when it was strengthened and weakened and, consequently, contributed to a highly biodiverse region with species adapted to historic temperature ranges [64]. Climate change projections suggest that this region is a warming hotspot where the rate of warming is in the top 10% globally [9], and the most extreme MHW on record was observed in this region with the 2011 Ningaloo Niño [3,65]. Despite some levels of adaptation to temperature shifts, the consequences of changes in benthic communities are expected to be profound [23]. Nevertheless, in this study, we found that deeper habitats were less affected by the discrete warming event in the 2010/11 MHW, where greater depths depict a more stable community with lower species turnover rates [66]. Yet, our understanding of the processes driving the community dynamics of deeper reefs is still in its infancy, as indicated by the large change we observed in community composition at 40 m sites of Rottnest Island in 2017, mainly driven by an increase in sponges which we were unable to associate with an environmental change or disturbance.

Despite the persistence of deep populations in deep habitats after the 2011 MHW, a single event is not enough to confirm the existence of deep refuges. The response of deep communities to future extreme events needs to be evaluated to confirm their role as a resilience mechanism for depth generalist species living in shallow reefs. This highlights the importance of continuous monitoring of benthic habitats at different depths. Furthermore, other ecological processes need to be evaluated across depth to confirm the existence of refuge at depth, such as changes to fecundity, transport of propagules from deep to shallow sites and post-settlement survival at shallower disturbed sites. Moreover, these processes may vary across species and the full definition of deep refuges may only apply to some species. For example, *S. dorycarpa* at 15 m depth at Rottnest Island did not show signs of recovery despite its persistence at deep sites. This may be related to its susceptibility to warm temperatures, which has shown to decrease settlement densities and post-settlement survival of germlings [67], or possibly to reduced fecundity with depth or unsuccessful transport of germlings from deep to shallow sites, all of these processes are currently unknown.

Macrophyte communities are quintessential to temperate reefs, providing valuable ecosystem services worth millions of dollars per year [68]. In Western Australia, the western rock lobster (which is endemic to this region) fishery alone is worth over AU\$300 M yr$^{-1}$ with numerous studies identifying *E. radiata* as critical habitat for adult lobsters [69]. Further research into deeper communities is required to fully understand their potential to act as refuges for shallow benthic foundation species and the ecosystem services they provide.

We suggest that deep benthic marine habitats in temperate Western Australia may play a role in buffering the impacts of a recent extreme MHW on the benthic communities found on the continental shelf and, therefore, have the potential to act as a refuge against future extreme climatic events potentially assisting the recovery of shallow-reef communities. If deep habitats are less affected by future extreme events, in the long term, they could act as refuges from climate change, and the range shifts in offshore reefs may be less extreme than projected for inshore systems. It is also essential that these offshore habitats are studied to assist resource managers, particularly in the planning of marine reserves and future proofing of fishery sustainability.

Data accessibility. Additional results supporting this article have been uploaded as the online electronic supplementary material. Datasets used in this study are publicly available online at https://figshare.com/articles/Benthic_codefile_Depth_moderates_loss_of_marine_benthic_communities_csv/7789538, for benthic classification code and data, and at https://figshare.com/articles/MARVL3Summer_

temperature_averages_and_anomalies-Abrolhos_Jurien_Rottnest/7789544, for temperature data.

**Authors' contributions.** G.A.K., R.K.H. and A.G.O. conceptualized the study and conducted fieldwork. G.A.K. and R.K.H. did an initial analysis on a subset of the data. A.G.O. conducted the complete data analysis. All authors contributed to interpreting the results and writing of the manuscript.

**Competing interests.** The authors declare that they have no competing interests.

**Funding.** G.A.K. and R.K.H. organized funding for AUV surveys (Marine Biodiversity NERP, Australian Research Council Grants (grant nos. DP150104251 and LE13010020) and the Integrated Marine Observing System (IMOS) through the Department of Innovation, Industry, Science and Research (DIISR), National Collaborative Research Infrastructure scheme). Fisheries Research and Development Corporation (FRDC) project no. 2008/013 helped cover costs for 2 of the years.

**Acknowledgements.** We acknowledge the WA Department of Primary Industries and Regional Development Habitat.

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
