## [Reviewer comments · Proceedings of the Royal Society B: Biological Sciences]

Review History

RSPB-2019-0894.R0 (Original submission)

Review form: Reviewer 1

Recommendation

Major revision is needed (please make suggestions in comments)

Scientific importance: Is the manuscript an original and important contribution to its field?

Excellent

General interest: Is the paper of sufficient general interest?

Excellent

Quality of the paper: Is the overall quality of the paper suitable?

Good

Is the length of the paper justified?

Yes

Should the paper be seen by a specialist statistical reviewer?

No

Do you have any concerns about statistical analyses in this paper? If so, please specify them explicitly in your report.

No

It is a condition of publication that authors make their supporting data, code and materials available - either as supplementary material or hosted in an external repository. Please rate, if applicable, the supporting data on the following criteria.

Is it accessible?

Yes

Is it clear?

Yes

Is it adequate?

Yes

Do you have any ethical concerns with this paper?

No

Comments to the Author

See attachment. (See Appendix A)

Review form: Reviewer 2

Recommendation

Major revision is needed (please make suggestions in comments)

Scientific importance: Is the manuscript an original and important contribution to its field?

Marginal

General interest: Is the paper of sufficient general interest?

Marginal

Quality of the paper: Is the overall quality of the paper suitable?

Marginal

Is the length of the paper justified?

Yes

Should the paper be seen by a specialist statistical reviewer?

No

Do you have any concerns about statistical analyses in this paper? If so, please specify them explicitly in your report.

Yes

It is a condition of publication that authors make their supporting data, code and materials available - either as supplementary material or hosted in an external repository. Please rate, if applicable, the supporting data on the following criteria.

Is it accessible?

Yes

Is it clear?

No

Is it adequate?

No

Do you have any ethical concerns with this paper?

No

Comments to the Author

General Comments

Giraldo-Ospina et al. provide an interesting set of data from biological transition reefs off of Western Australia. This data of remote surveys of benthic cover at 15, 25, and 40 m water depths across the 2011 marine heat wave is combined with opportunistic temperature data to suggest that benthic sub-tropical to temperate habitats in their region are less sensitive to thermal anomalies and are therefore represent a refugium from thermal stress. While the paper presents an interesting set of data (albeit with some errors), I do not find the argument for a depth refugium very convincing. Many of the benthic changes in shallow water seem very slight to me. This moderate change was indicated by the authors in reference to more severe changes that occurred in even shallower areas in 2011. If their attempt was to say that anything 15+ m is the depth refuge, then it was not compared well with shallow water data. If their attempt was to say the 40 m relative to 15 and 25 m was a refuge, then the data don't seem to show a strong negative response for the shallower sampling depths. To me, these changes could be natural wobbles of benthic communities (particularly faster growing macroalgae) than a response to a marine heat wave. Also, the temperature data don't suggest a strong difference in the summer temperatures nor the anomalies in the 2011 heat wave between the depths. What then is the mechanism for a refuge? Is it, as they note, that deeper areas show more high anomalies in non heat wave years? The difference in anomalies at any summer period across the depths appears very slight, even if the 40 m has a tendency for higher anomalies. If they posit this as a mechanism then I think more due diligence is required to make that case.

On another note, even if the authors were able to show that deeper areas convincingly avoided change due to high temperature in 2011, then this demonstrates a one off phenomena (refuge) not a long-term escape from this type of stress that protects biodiversity (refugium). To begin to demonstrate a refugium they would need additional data. That's not to say it isn't useful to talk about a habitat refuge and their characteristics, but there is work to be done to make a convincing case for a heat wave refugium.

Specific Comments

Line 51-53: Is this more specific to all refugia or refugia in BTZ's? The set up would seem like BTZ's, but it seems like a more general statement. The flow seems to be about BTZ's and it comes off a bit awkward that they would be the justification for identifying refugia.

Para beginning Line 54: If marine heat waves are also involved in the process of coral bleaching, which the originators of the concept state (<http://www.marineheatwaves.org/all-about-mhws.html>), and this is one of the most pressing biodiversity crises globally, seems like it deserves a mention in this paragraph.

Line 74: "the bleached coral was reported to be 12-100 %". What do you mean specifically? The individual coral colonies has 12-100% of living tissue stark white bleached? Please be more specific.

Line 85-86: I would avoid bringing up other reasons that deeper habitats might be buffered, such

as local anthro pollution, as it just detracts from the point you are making about depth refugia from thermal stress and ecological change.

Line 83: I would say "may benefit" because it is not necessarily true in all systems across all events. E.g., Smith TB, Gyory J, Brandt ME, Miller WJ, Jossart J, Nemeth RS (2016) Caribbean mesophotic coral ecosystems are unlikely climate change refugia. *Global Change Biology* 22:2756–2765

Lines 101-105: First, this is a long sentence. Second, you cannot argue that deep benthic habitats (as in all habitats) are a refugia based on a single study (one event) from Western Australia? Rephrase this to make it more specific to the habitat where your study was conducted. Also, a refugium can only be determined over multiple events. A refuge can be determined for one event. So, you are going to have hard task making a convincing argument that this represents a refugium without more evidence over more events.

Line 119: Change to "transects of the seafloor"

The paragraphs starting on line 205 that introduce the site changes over time have numerous errors in reference to the supplementary material. Below are a couple of examples, but nearly all references to the supplemental figures are incorrect.

Line 208: The Fig. S2 presents no information on coral bleaching. Also, the figure only lists Foliose coral cover. Is this the only morpho-type that was present? What about the massive, encrusting, and staghorn/tabulate presented in Fig. 2?

Line 209: The turf figure is S1 in the supplemental, not S3

For both coral and turf you use the term "significant" implying a statistical test, but I do not see an associated statistical test aside from the general PERMANOVA. The PERMANOVA is showing the overall depth*time trend individually at each location. It is not testing specifically for an increase or decrease in any time period, which is what your statement says is "significant". The use the term "trend" in introductory sentence (Line 205), so perhaps this is not statistical, in which case the term "significant" should be avoided. Alternatively, you could probably construct tests of change over time to support you contention of an increase (e.g., a significant increase in turf at Abrolhos after the 2011 marine heatwave).

For sand, how does including an abiotic variable that should not respond to a marine heatwave affect the rest of the data in the PCO?

Figure 4c. Is the data point for Abrolhos 40m hidden in 2011 or not there? You should not in the caption.

Line 225-226: "At the 15 m sites there was an increase in *E. radiata* (figure 4a)". This doesn't look like much of a change to me. It is especially unconvincing without a statistical comparison.

Line 237: Use past tense "ranged"

Figure 5. In cases where there is no data point visible (either behind another data point or there was no data) please indicate in the caption if there is a data point and where it is. E.g., Abrohos 15 and 25 m in 1995 are overlapping as indicated by Table S3, but you would have to dig into the ESM to figure that out.

Figure 5b. A 0 line would be helpful for reference.

Inconsistency in the format of literature citations in the References section. There also seem to be notes to finalize references (e.g., #45 "double check")

Decision letter (RSPB-2019-0894.R0)

17-May-2019

Dear Mrs Giraldo Ospina:

I am writing to inform you that your manuscript RSPB-2019-0894 entitled "Deep refugia moderate loss of shallow marine foundation species after an extreme marine heatwave" has, in its current form, been rejected for publication in *Proceedings B*.

This action has been taken on the advice of referees, who have recommended that substantial revisions are necessary. With this in mind we would be happy to consider a resubmission, provided the comments of the referees are fully addressed. However please note that this is not a provisional acceptance.

Sincerely,
Proceedings B
mailto:proceedingsb@royalsociety.org

Associate Editor
Board Member: 1
Comments to Author:

Two expert reviewers have commented on this manuscript on the possible role of marine depths as refugia from marine heatwaves. Whilst they find the results interesting, both present comprehensive sets of questions and recommendations about the work. Importantly, both reviewers request greater clarity in the use of the term "refugia", or clearer evidence to support the claims that refugia are observed in this system.

Reviewer(s)' Comments to Author:

Referee: 1

Comments to the Author(s)
See attachment

Referee: 2

Comments to the Author(s)
General Comments

Giraldo-Ospina et al. provide an interesting set of data from biological transition reefs off of Western Australia. This data of remote surveys of benthic cover at 15, 25, and 40 m water depths across the 2011 marine heat wave is combined with opportunistic temperature data to suggest

that benthic sub-tropical to temperate habitats in their region are less sensitive to thermal anomalies and are therefore represent a refugium from thermal stress. While the paper presents an interesting set of data (albeit with some errors), I do not find the argument for a depth refugium very convincing. Many of the benthic changes in shallow water seem very slight to me. This moderate change was indicated by the authors in reference to more severe changes that occurred in even shallower areas in 2011. If their attempt was to say that anything 15+ m is the depth refuge, then it was not compared well with shallow water data. If their attempt was to say the 40 m relative to 15 and 25 m was a refuge, then the data don't seem to show a strong negative response for the shallower sampling depths. To me, these changes could be natural wobbles of benthic communities (particularly faster growing macroalgae) than a response to a marine heat wave. Also, the temperature data don't suggest a strong difference in the summer temperatures nor the anomalies in the 2011 heat wave between the depths. What then is the mechanism for a refuge? Is it, as they note, that deeper areas show more high anomalies in non heat wave years? The difference in anomalies at any summer period across the depths appears very slight, even if the 40 m has a tendency for higher anomalies. If they posit this as a mechanism then I think more due diligence is required to make that case.

On another note, even if the authors were able to show that deeper areas convincingly avoided change due to high temperature in 2011, then this demonstrates a one off phenomena (refuge) not a long-term escape from this type of stress that protects biodiversity (refugium). To begin to demonstrate a refugium they would need additional data. That's not to say it isn't useful to talk about a habitat refuge and their characteristics, but there is work to be done to make a convincing case for a heat wave refugium.

Specific Comments

Line 51-53: Is this more specific to all refugia or refugia in BTZ's? The set up would seem like BTZ's, but it seems like a more general statement. The flow seems to be about BTZ's and it comes off a bit awkward that they would be the justification for identifying refugia.

Para beginning Line 54: If marine heat waves are also involved in the process of coral bleaching, which the originators of the concept state (<http://www.marineheatwaves.org/all-about-mhws.html>), and this is one of the most pressing biodiversity crises globally, seems like it deserves a mention in this paragraph.

Line 74: "the bleached coral was reported to be 12-100 %". What do you mean specifically? The individual coral colonies has 12-100% of living tissue stark white bleached? Please be more specific.

Line 85-86: I would avoid bringing up other reasons that deeper habitats might be buffered, such as local anthro pollution, as it just detracts from the point you are making about depth refugia from thermal stress and ecological change.

Line 83: I would say "may benefit" because it is not necessarily true in all systems across all events. E.g., Smith TB, Gyory J, Brandt ME, Miller WJ, Jossart J, Nemeth RS (2016) Caribbean mesophotic coral ecosystems are unlikely climate change refugia. *Global Change Biology* 22:2756-2765

Lines 101-105: First, this is a long sentence. Second, you cannot argue that deep benthic habitats (as in all habitats) are a refugia based on a single study (one event) from Western Australia? Rephrase this to make it more specific to the habitat where your study was conducted. Also, a refugium can only be determined over multiple events. A refuge can be determined for one event. So, you are going to have hard task making a convincing argument that this represents a refugium without more evidence over more events.

Line 119: Change to "transects of the seafloor"

The paragraphs starting on line 205 that introduce the site changes over time have numerous errors in reference to the supplementary material. Below are a couple of examples, but nearly all references to the supplemental figures are incorrect.

Line 208: The Fig. S2 presents no information on coral bleaching. Also, the figure only lists Foliose coral cover. Is this the only morpho-type that was present? What about the massive, encrusting, and staghorn/tabulate presented in Fig. 2?

Line 209: The turf figure is S1 in the supplemental, not S3

For both coral and turf you use the term “significant” implying a statistical test, but I do not see an associated statistical test aside from the general PERMANOVA. The PERMANOVA is showing the overall depth*time trend individually at each location. It is not testing specifically for an increase or decrease in any time period, which is what your statement says is “significant”. The use the term “trend” in introductory sentence (Line 205), so perhaps this is not statistical, in which case the term “significant” should be avoided. Alternatively, you could probably construct tests of change over time to support you contention of an increase (e.g., a significant increase in turf at Abrolhos after the 2011 marine heatwave).

For sand, how does including an abiotic variable that should not respond to a marine heatwave affect the rest of the data in the PCO?

Figure 4c. Is the data point for Abrolhos 40m hidden in 2011 or not there? You should not in the caption.

Line 225-226: “At the 15 m sites there was an increase in *E. radiata* (figure 4a)”. This doesn’t look like much of a change to me. It is especially unconvincing without a statistical comparison.

Line 237: Use past tense “ranged”

Figure 5. In cases where there is no data point visible (either behind another data point or there was no data) please indicate in the caption if there is a data point and where it is. E.g., Abrohos 15 and 25 m in 1995 are overlapping as indicated by Table S3, but you would have to dig into the ESM to figure that out.

Figure 5b. A 0 line would be helpful for reference.

Inconsistency in the format of literature citations in the References section. There also seem to be notes to finalize references (e.g., #45 “double check”)

Author's Response to Decision Letter for (RSPB-2019-0894.R0)

See Appendix B.

RSPB-2019-2625.R0

Review form: Reviewer 3

Recommendation

Reject – article is scientifically unsound

Scientific importance: Is the manuscript an original and important contribution to its field?

Acceptable

General interest: Is the paper of sufficient general interest?

Acceptable

Quality of the paper: Is the overall quality of the paper suitable?

Marginal

Is the length of the paper justified?

Yes

Should the paper be seen by a specialist statistical reviewer?

No

Do you have any concerns about statistical analyses in this paper? If so, please specify them explicitly in your report.

No

It is a condition of publication that authors make their supporting data, code and materials available - either as supplementary material or hosted in an external repository. Please rate, if applicable, the supporting data on the following criteria.

Is it accessible?

Yes

Is it clear?

No

Is it adequate?

No

Do you have any ethical concerns with this paper?

No

Comments to the Author

General comments.

This paper uses video transect data to evaluate the response of reef communities to a large marine heatwave across a depth and latitudinal gradient in Western Australia. It is quite a remarkable and unique dataset that required substantial work to collect and analyze. I commend the authors for their efforts in this regard. As much as I liked the data I was less enamored with the interpretation and conclusions that the authors derived from them and by the strong focus on the potential role of deep reefs as refugia for species on shallow reefs in which the paper is cast.

The primary focus of the results is on the statistical significance of multivariate analyses used to show differences in the trajectories of deep and shallow reef communities during an 8-year period that encompasses one year before and 6 years after the 2011 heatwave. The authors also described directional trends in key species and taxonomic groups without providing formal analyses of these trends. Conspicuously absent is any mention of the magnitude of change in key species over time and among depths. Such information is critical for evaluating the degree to which deep reefs served as a refuge from the heatwave, and the likelihood that they may serve as refugia for populations of shallow reef species severely damaged or extirpated by the heatwave. While the results of multivariate analyses are useful for showing the communities at the three depths differed and displayed different temporal trajectories, they do not provide convincing evidence for the role of deep reefs as a refuge or refugium.

Habitat forming foundation species are featured prominently in the Introduction and Discussion when discussing the importance of deep refugia in facilitating the recovery of shallow reef populations. Unfortunately, the foundation species along this latitudinal gradient were never clearly identified. The percent cover values presented in Figure 4 and the supplemental figures indicate that the only species that seemed abundant enough to rise to the level of a habitat forming foundation species was *Ecklonia*. However, this species was only abundant at one site (Rottneest) and at this site it did not show a marked response to the heatwave at any depth. Thus no evidence was presented that depth served as a refuge for this species or that it has a potential to serve as a refugium.

In my view one of the more interesting results was the contrast between the response of *Ecklonia* to the heatwave in this study compared to that published in other studies. This result did not receive the attention that I think it deserved, which I thought was unfortunate.

In sum, I think the data presented in this paper are quite good and the results are interesting. However, I thought the author's interpretation of the data was inadequate and at times misleading, and that many of their conclusions (especially those pertaining to the role of deep reefs as refugia) were not supported by the results of their study.

Specific comments

Title - Suggest dropping the second clause from the title as the paper does not answer the question of whether deep reefs act as climate refugia.

Lines 47 and 92. Might want to cite Ladah and Zertuche-Gonzalez 2007 (*Botanica Marina* 47: 367-372) since they argued deep reefs served as a refuge for the giant kelp in Mexico during the 1997-98 El Niño.

Lines 158-167. Might consider looking at metrics other than the deviation from the mean (e.g. maximum, or 90th percentile) given that many species exhibit a threshold response to temperature stress.

Line 197. Incomplete sentence. Not possible to evaluate this statement.

Line 212 Fig S5. Confusing statement as the increase in *Sargassum* in Fig. S5 was relatively small at 25m (as was *Ecklonia*), while the increase in *Sargassum* cover was comparatively higher at 15 m. Need to clarify.

Line 214 -215. The minimal change observed at the Abrolhos 40 m site likely reflects the fact that the cover of most species and taxonomic groups was really low at Abrolhos 40 m (near zero for most groups other than branched red algae and sponges). This needs to be reported otherwise the statement seems to misrepresent the patterns observed.

Line 222. Fig 4c shows trends in fine branching red algae, but the text describes increases in "coarse branching algae". Need to clarify.

Line 220. Figure 4c shows encrusting red algae. Should the reference be Figure 4d which shows fine branching red algae? Regardless, there is little fine branching red algae at any depth in any year at Rottneest according to Fig. 4, so it's hard to make much of this statement.

Line 230-231. The emphasis on recovery in community composition here seems a bit misleading given that Figure 4 shows that the % cover of *Ecklonia* and encrusting red algae (which formed the majority of the cover at shallow depths at Rottneest) remained relatively constant during the time series. Consider revising to more accurately describe the changes and/or lack of change observed.

Lines 235-236. Need to provide a more detailed description of the magnitude of these changes (e.g. sponges increased from ~ 4% in 2010 to ~ 9% in 2017).

Lines 236-238. Again, it does not seem sufficient to simply state that *S. dorycarpa* declined. The reader needs to know how much it declined. In this case it looks like average cover of *S. dorycarpa* varied between about 3 and 7% at 40 m during the time series. The details matter because you are arguing for importance of deep reefs in providing refuge during a heatwave and in serving as refugia for the recovery populations in shallower areas that were decimated by the heatwave.

Line 2560-262. need to be more clear about what the evidence is that you are referring to. The specific taxa need to be identified.

Lines 265-269. The purported depth buffering is not evident from this study. Catastrophic loss was only shown for *S. doryocarpa* and this species was largely absent or rare at the two northern sites at all depths. In the case of *Ecklonia* there seemed to be little difference in temporal response to heatwave among the three depths at Rottneest where it was most abundant. It was never present shallow at Jurien, or deep at Abrolhos.

Lines 273-277. This is another instance of where information on the magnitude of change as well as the direction is needed to provide context for evaluating ecological significance.

Lines 278-281. This statement seems a bit misleading as *Scytothalia* was absent or rare at all depths at Abrolhos and Jurien and fine branching red algae were rare or absent at Rottneest. Perhaps my criticism in this case is due to my confusion about what "transition zone" refers to in this case. In any case, this statement needs to be revised for clarity and accuracy.

Lines 281-285. The size of a source population as well as its fecundity and physical connectivity is important in determining its ability to serve as a refugia, which in this paper is defined as a "region that "that facilitate the temporal and spatial resilience of biological communities and enable the recovery from biophysical disturbances". The argument that deep reef populations may be key drivers of shallow reef resilience would be much stronger if the reader was provided with some information on the sizes of the populations of interest (habitat forming species? other species?)

Lines 291-292. The finding that the results of this study differed from those of Wernberg et al is quite interesting given the notoriety of these earlier studies. It would be worth expanding on why the results of the two studies likely differed. Is it because different sites were sampled? Same sites but different depths? Any additional information that can be provide on this discrepancy would be useful helping to understand the nature and extent of impacts resulting from this heatwave

Line 315. Insert "positive" before net photosynthesis.

Line 333. These references are a bit data and the author's might want to check this statement for accuracy. By most accounts the 2014-15 "Blob" and El Nino in the central and eastern Pacific seemed was more extreme than the 2011 heatwave in WA.

Line 342-343. Determining the role of deep reefs as a "resilience mechanism for shallow species" not only requires documenting the response of deep reefs, but it also requires documenting the recovery of shallow reefs and the genetic or demographic connectivity evidence that shows source populations on deep reefs contributed to this recovery.

Review form: Reviewer 4 (Joaquim Garrabou)

Recommendation

Accept with minor revision (please list in comments)

Scientific importance: Is the manuscript an original and important contribution to its field?

Good

General interest: Is the paper of sufficient general interest?

Excellent

Quality of the paper: Is the overall quality of the paper suitable?

Good

Is the length of the paper justified?

Yes

Should the paper be seen by a specialist statistical reviewer?

No

Do you have any concerns about statistical analyses in this paper? If so, please specify them explicitly in your report.

No

It is a condition of publication that authors make their supporting data, code and materials available - either as supplementary material or hosted in an external repository. Please rate, if applicable, the supporting data on the following criteria.

Is it accessible?

Yes

Is it clear?

Yes

Is it adequate?

Yes

Do you have any ethical concerns with this paper?

No

Comments to the Author

Please see my comments in the attached file. (See Appendix C)

Decision letter (RSPB-2019-2625.R0)

17-Dec-2019

Dear Mrs Giraldo Ospina:

I am writing to inform you that we have now obtained responses from referees on manuscript RSPB-2019-2625 entitled "Depth moderates loss of marine foundation species after an extreme marine heatwave: Will deep temperate reefs act as climate refugia?" which you submitted to Proceedings B.

Unfortunately, your manuscript has been rejected following full peer review. Competition for space in Proceedings B is currently extremely severe, as many more manuscripts are submitted to us than we have space to print. We are therefore only able to publish those that are exceptional, convincing and present significant advances of broad interest, and must reject many good manuscripts.

Please find below the comments received from referees concerning your manuscript, not including confidential reports to the Editor. I hope you may find these useful should you wish to submit your manuscript elsewhere.

We are sorry that your manuscript has had an unfavourable outcome, but would like to thank you for offering your work to Proceedings B.

Sincerely,
Dr Daniel Costa
mailto: proceedingsb@royalsociety.org

Associate Editor
Board Member
Comments to Author:

The resubmission of this work on the role of depth in the responses of reefs to marine heatwaves has been seen by two new expert referees. Whilst both referees saw considerable merit in the dataset and the research conducted, both also considered there to be some shortcomings in presentation and interpretation - referee 3 in particular raising a number of important concerns which would need to be carefully and comprehensively addressed by the authors.

Reviewer(s)' Comments to Author:

Referee: 3

Comments to the Author(s).
General comments.

This paper uses video transect data to evaluate the response of reef communities to a large marine heatwave across a depth and latitudinal gradient in Western Australia. It is quite a remarkable and unique dataset that required substantial work to collect and analyze. I commend the authors for their efforts in this regard. As much as I liked the data I was less enamored with the interpretation and conclusions that the authors derived from them and by the strong focus on the potential role of deep reefs as refugia for species on shallow reefs in which the paper is cast.

The primary focus of the results is on the statistical significance of multivariate analyses used to show differences in the trajectories of deep and shallow reef communities during an 8-year period that encompasses one year before and 6 years after the 2011 heatwave. The authors also described directional trends in key species and taxonomic groups without providing formal analyses of these trends. Conspicuously absent is any mention of the magnitude of change in key species over time and among depths. Such information is critical for evaluating the degree to which deep reefs served as a refuge from the heatwave, and the likelihood that they may serve as refugia for populations of shallow reef species severely damaged or extirpated by the heatwave. While the results of multivariate analyses are useful for showing the communities at the three depths differed and displayed different temporal trajectories, they do not provide convincing evidence for the role of deep reefs as a refuge or refugium.

Habitat forming foundation species are featured prominently in the Introduction and Discussion when discussing the importance of deep refugia in facilitating the recovery of shallow reef populations. Unfortunately, the foundation species along this latitudinal gradient were never clearly identified. The percent cover values presented in Figure 4 and the supplemental figures indicate that the only species that seemed abundant enough to rise to the level of a habitat forming foundation species was *Ecklonia*. However, this species was only abundant at one site (Rottnest) and at this site it did not show a marked response to the heatwave at any depth. Thus no evidence was presented that depth served as a refuge for this species or that it has a potential to serve as a refugium.

In my view one of the more interesting results was the contrast between the response of *Ecklonia* to the heatwave in this study compared to that published in other studies. This result did not receive the attention that I think it deserved, which I thought was unfortunate.

In sum, I think the data presented in this paper are quite good and the results are interesting. However, I thought the author's interpretation of the data was inadequate and at times misleading, and that many of their conclusions (especially those pertaining to the role of deep reefs as refugia) were not supported by the results of their study.

Specific comments

Title - Suggest dropping the second clause from the title as the paper does not answer the question of whether deep reefs act as climate refugia.

Lines 47 and 92. Might want to cite Ladah and Zertuche-Gonzalez 2007 (*Botanica Marina* 47: 367–372) since they argued deep reefs served as a refuge for the giant kelp in Mexico during the 1997–98 El Niño.

Lines 158–167. Might consider looking at metrics other than the deviation from the mean (e.g. maximum, or 90th percentile) given that many species exhibit a threshold response to temperature stress.

Line 197. Incomplete sentence. Not possible to evaluate this statement.

Line 212 Fig S5. Confusing statement as the increase in *Sargassum* in Fig. S5 was relatively small at 25m (as was *Ecklonia*), while the increase in *Sargassum* cover was comparatively higher at 15 m. Need to clarify.

Line 214 –215. The minimal change observed at the Abrolhos 40 m site likely reflects the fact that the cover of most species and taxonomic groups was really low at Abrolhos 40 m (near zero for most groups other than branched red algae and sponges). This needs to be reported otherwise the statement seems to misrepresent the patterns observed.

Line 222. Fig 4c shows trends in fine branching red algae, but the text describes increases in “coarse branching algae”. Need to clarify.

Line 220. Figure 4c shows encrusting red algae. Should the reference be Figure 4d which shows fine branching red algae? Regardless, there is little fine branching red algae at any depth in any year at Rottneest according to Fig. 4, so it's hard to make much of this statement.

Line 230–231. The emphasis on recovery in community composition here seems a bit misleading given that Figure 4 shows that the % cover of *Ecklonia* and encrusting red algae (which formed the majority of the cover at shallow depths at Rottneest) remained relatively constant during the time series. Consider revising to more accurately describe the changes and/or lack of change observed.

Lines 235–236. Need to provide a more detailed description of the magnitude of these changes (e.g. sponges increased from ~ 4% in 2010 to ~ 9% in 2017).

Lines 236–238. Again, it does not seem sufficient to simply state that *S. dorycarpa* declined. The reader needs to know how much it declined. In this case it looks like average cover of *S. dorycarpa* varied between about 3 and 7% at 40 m during the time series. The details matter because you are arguing for importance of deep reefs in providing refuge during a heatwave and in serving as refugia for the recovery populations in shallower areas that were decimated by the heatwave.

Line 2560–262. need to be more clear about what the evidence is that you are referring to. The specific taxa need to be identified.

Lines 265–269. The purported depth buffering is not evident from this study. Catastrophic loss

was only shown for *S. doryocarpa* and this species was largely absent or rare at the two northern sites at all depths. In the case of *Ecklonia* there seemed to be little difference in temporal response to heatwave among the three depths at Rottneest where it was most abundant. It was never present shallow at Jurien, or deep at Abrolhos.

Lines 273-277. This is another instance of where information on the magnitude of change as well as the direction is needed to provide context for evaluating ecological significance.

Lines 278-281. This statement seems a bit misleading as *Scytothalia* was absent or rare at all depths at Abrolhos and Jurien and fine branching red algae were rare or absent at Rottneest. Perhaps my criticism in this case is due to my confusion about what "transition zone" refers to in this case. In any case, this statement needs to be revised for clarity and accuracy.

Lines 281-285. The size of a source population as well as its fecundity and physical connectivity is important in determining its ability to serve as a refugia, which in this paper is defined as a "region that "that facilitate the temporal and spatial resilience of biological communities and enable the recovery from biophysical disturbances". The argument that deep reef populations may be key drivers of shallow reef resilience would be much stronger if the reader was provided with some information on the sizes of the populations of interest (habitat forming species? other species?)

Lines 291-292. The finding that the results of this study differed from those of Wernberg et al is quite interesting given the notoriety of these earlier studies. It would be worth expanding on why the results of the two studies likely differed. Is it because different sites were sampled? Same sites but different depths? Any additional information that can be provide on this discrepancy would be useful helping to understand the nature and extent of impacts resulting from this heatwave

Line 315. Insert "positive" before net photosynthesis.

Line 333. These references are a bit data and the author's might want to check this statement for accuracy. By most accounts the 2014-15 "Blob" and El Nino in the central and eastern Pacific seemed was more extreme than the 2011 heatwave in WA.

Line 342-343. Determining the role of deep reefs as a "resilience mechanism for shallow species" not only requires documenting the response of deep reefs, but it also requires documenting the recovery of shallow reefs and the genetic or demographic connectivity evidence that shows source populations on deep reefs contributed to this recovery.

Referee: 4

Comments to the Author(s).

Please see my comments in the attached file

Author's Response to Decision Letter for (RSPB-2019-2625.R0)

See Appendix D.

RSPB-2020-0709.R0

Review form: Reviewer 3

Recommendation

Accept with minor revision (please list in comments)

Scientific importance: Is the manuscript an original and important contribution to its field?

Good

General interest: Is the paper of sufficient general interest?

Good

Quality of the paper: Is the overall quality of the paper suitable?

Good

Is the length of the paper justified?

Yes

Should the paper be seen by a specialist statistical reviewer?

No

Do you have any concerns about statistical analyses in this paper? If so, please specify them explicitly in your report.

Yes

It is a condition of publication that authors make their supporting data, code and materials available - either as supplementary material or hosted in an external repository. Please rate, if applicable, the supporting data on the following criteria.

Is it accessible?

No

Is it clear?

No

Is it adequate?

No

Do you have any ethical concerns with this paper?

No

Comments to the Author

See attached file Proc_B_2020-0709_review.pdf. (See Appendix E)

Decision letter (RSPB-2020-0709.R0)

14-Apr-2020

Dear Mrs Giraldo Ospina:

Your manuscript has now been peer reviewed and the reviews have been assessed by an Associate Editor. The reviewers' comments (not including confidential comments to the Editor) and the comments from the Associate Editor are included at the end of this email for your reference. As you will see, the reviewers and the Editors have raised some concerns with your manuscript and we would like to invite you to revise your manuscript to address them.

Research ethics:

Use of animals and field studies:

Please submit a copy of your revised paper within three weeks. If we do not hear from you within this time your manuscript will be rejected. If you are unable to meet this deadline please let us know as soon as possible, as we may be able to grant a short extension.

Best wishes,
Dr Daniel Costa
mailto:proceedingsb@royalsociety.org

Associate Editor Board Member

Comments to Author:

The comprehensive revisions made by the authors were welcomed by the referee, who felt that the manuscript had been much improved. The referee provides a helpful set of comments on remaining areas where the presentation of the manuscript could be strengthened, including a recommendation for a modification of the Permanova analysis to distinguish between normal interannual variability and that caused by the marine heatwave, and a comment regarding the temperature data presented. Please also ensure that data uploaded with the paper include any necessary metadata or supporting information, or links to raw data sources.

Reviewer(s)' Comments to Author:

Referee: 3

Comments to the Author(s).

See attached file Proc_B_2020-0709_review.pdf

Author's Response to Decision Letter for (RSPB-2020-0709.R0)

See Appendix F.

Decision letter (RSPB-2020-0709.R1)

05-May-2020

Dear Mrs Giraldo Ospina

I am pleased to inform you that your Review manuscript RSPB-2020-0709.R1 entitled "Depth moderates loss of marine foundation species after an extreme marine heatwave: could deep temperate reefs act as a refuge?" has been accepted for publication in Proceedings B.

The referee(s) do not recommend any further changes. Therefore, please proof-read your manuscript carefully and upload your final files for publication. Because the schedule for publication is very tight, it is a condition of publication that you submit the revised version of your manuscript within 7 days. If you do not think you will be able to meet this date please let me know immediately.

To upload your manuscript, log into <http://mc.manuscriptcentral.com/prsb> and enter your Author Centre, where you will find your manuscript title listed under "Manuscripts with Decisions." Under "Actions," click on "Create a Revision." Your manuscript number has been appended to denote a revision.

You will be unable to make your revisions on the originally submitted version of the manuscript. Instead, upload a new version through your Author Centre.

1) A text file of the manuscript (doc, txt, rtf or tex), including the references, tables (including captions) and figure captions. Please remove any tracked changes from the text before submission. PDF files are not an accepted format for the "Main Document".

2) A separate electronic file of each figure (tiff, EPS or print-quality PDF preferred). The format should be produced directly from original creation package, or original software format. Please note that PowerPoint files are not accepted.

3) Electronic supplementary material: this should be contained in a separate file from the main text and the file name should contain the author's name and journal name, e.g. `authorname_procb_ESM_figures.pdf`

All supplementary materials accompanying an accepted article will be treated as in their final form. They will be published alongside the paper on the journal website and posted on the online figshare repository. Files on figshare will be made available approximately one week before the accompanying article so that the supplementary material can be attributed a unique DOI. Please see: <https://royalsociety.org/journals/authors/author-guidelines/>

4) Data-Sharing and data citation

It is a condition of publication that data supporting your paper are made available. Data should be made available either in the electronic supplementary material or through an appropriate repository. Details of how to access data should be included in your paper. Please see <https://royalsociety.org/journals/ethics-policies/data-sharing-mining/> for more details.

If you wish to submit your data to Dryad (<http://datadryad.org/>) and have not already done so you can submit your data via this link <http://datadryad.org/submit?journalID=RSPB&manu=RSPB-2020-0709.R1> which will take you to your unique entry in the Dryad repository.

Once again, thank you for submitting your manuscript to Proceedings B and I look forward to receiving your final version. If you have any questions at all, please do not hesitate to get in touch.

Sincerely,
Dr Daniel Costa
Editor, Proceedings B
<mailto:proceedingsb@royalsociety.org>

Associate Editor Board Member

Comments to Author:

Thank you to the authors for carefully addressing all of the remaining review comments in their revision and response to referees. My few remaining comments refer to minor grammatical issues:

There are a few places where "where" is used instead of "were" – please correct these on lines 184, 258, 276

I would remove the commas after species or place names on lines 26, 236, 299, and 311, and after "Although" on line 300.

Line 80 – I think this should be "tropical herbivorous fishes"

Line 110 – "foundation species"

Line 136 – "locations"

Line 198 – there is some repetition in this sentence. You can delete "were observed" and keep "were evident"

Line 239 – "it changed" to replace "it show change"

Line 256 – "cover of E. radiata"

Line 259 – "turf cover increased by around 10%"

Lines 288-290 – the structure of this sentence needs modifying to make the point clear. Maybe you can write "supporting our hypothesis that deep water refuges exhibit a potential buffering effect from extreme climatic events, that allows the persistence of kelp dominated communities".

Line 295 – "mixed assemblages"

Line 331 – remove "of" from "lacked enough"

Line 359 – "consequences ... are expected" (not "is")

Line 620 (Fig 4 legend) – "colours describe an increase"

Decision letter (RSPB-2020-0709.R2)

07-May-2020

Dear Mrs Giraldo Ospina

I am pleased to inform you that your manuscript entitled "Depth moderates loss of marine foundation species after an extreme marine heatwave: could deep temperate reefs act as a refuge?" has been accepted for publication in Proceedings B.

You can expect to receive a proof of your article from our Production office in due course, please check your spam filter if you do not receive it. PLEASE NOTE: you will be given the exact page

length of your paper which may be different from the estimation from Editorial and you may be asked to reduce your paper if it goes over the 10 page limit.

Open Access

Paper charges

Sincerely,

Appendix A

Review Giraldo-Ospina et al

Marine heatwaves are being seen as increasingly important agents of change linked with rising sea temperatures. The paper presented here represents an interesting piece of research making use of a 6-year (2010-2017) time series of AUV images across three regions and three depth horizons in Western Australia which were subjected to a large marine heatwave in the Austral summer of 2011. The authors make the case that because deeper water communities seemed less impacted by the 2011 marine heatwave that they could act as a refugia for shallow water communities. The fact that deeper water communities seemed to be less impacted than shallow water communities is clear to see in the data. I am more sceptical with the univariate analyses which seem to indicate idiosyncratic responses across species and locations and believe that the authors need to be less general in their description of this data in the results.

While I can appreciate the authors coming from the refugia angle, I am also not sure that the data support the idea of a refugia. For example, *Scytothalia* abundance holds up fairly well in the deeper waters of Rottnest island, but sees significant drops in shallower waters during the heatwave. The decrease in shallow waters continues. If deeper waters were acting as a refuge would the authors not expect some recovery from these deeper water populations. This is only one example from data presented, but none of the data presented really demonstrates that deeper waters are acting as a refugia to allow recovery to shallower waters. I believe this warrants some discussion.

Overall I think this is generally a well written ms on an important and interesting topic with a novel angle. I believe that this ms will be well read and cited, but feel that the results need to be clearer and the discussion to be developed in more depth.

Specific comments

Line 22 suggest adding 'after the heatwave' following recovery

Line 35 Delete refugia as glacial refugia are not just found for forest species, but a range of species in terrestrial and marine habitats.

Line 44 Suggest Smale et al 2019 is a better reference here as it focuses on the biological responses rather than how marine heatwaves have changed

Line 55 Frolicher et al 2018 Nature 560: 360-364 is a better reference about future MHWs as this ms uses climate models to predict the future whereas Oliver is more focused over the past 100 years.

Line 64 Suggest being a little more specific about the mechanism when suggesting impacts will depend on location and species affected. I would suggest including factors such as position in thermal range, potential for local thermal adaptation, prevalent current, dispersal ability etc.

Line 70-79 Given that the paper focuses a lot on canopy forming algae I suggest the authors refer to some of the impacts to e.g. kelp communities. Perhaps also worth mentioning tropicalisation of marine fauna leading to altered herbivory.

Line 85 Suggest you can delete 'effect of'

Line 119 Add 'across' after transects

Line 172 Should the first mention of site in this line actually be location?

Line 178-188 I found this section a little confusing to read and also some important information was missing. It seems to me that the two paragraphs can be combined as they appear to be explaining the same type of analysis. The confusing aspect relate to the fact the data seems to have been treated differently e.g. some used Euclidean distance (appropriate for univariate PERMANOVA) and some used Bray-Curtis resemblance matrix. Some clarity here would be helpful. Also the authors need to state with model they used to analyse the data (this would be different for the multivariate and univariate analyses) and also how many permutations were undertaken to determine significance. The authors also need to state why a dummy variable was added.

Line 195 I think the authors need to make sure the reader is clear they are discussing shallow waters here as Fig 2b does not suggest this assemblage for deeper waters. Also suggest being specific here and saying Sargassum to avoid confusion with other large macroalgae that are discussed further into the paragraph.

Line 198 Fig 2b doesn't indicate that Ecklonia is important. I am that it was present so perhaps worth stating something about drivers of community structure.

Line 199 I don't agree that fine-branching red algae seems to be a driver for deeper Abrolhos sites and Jurien Bay. The only driver seems to be sand. Reference to percent cover sand should be Fig S5.

Line 204 See comment above

Line 208 Fig S2 doesn't show anything about bleached coral. It states how the percent cover of coral changes. Suggest rewording.

Line 209 Throughout the SOM the figures are not in the order stated in the text. Please make sure this is corrected.

Line 210 Reference to foliose coral should be S2. Also make clear that you are referring to shallow benthic communities in the next sentence.

Line 211-214 Suggest the authors make it clear that it appears that these species have not recovered since the heatwave. Also S5 should be S3 and S6 should be S4.

Line 219 Add 'at 15m' after recovery

Line 220 S6 should be S4

Lines 216-222 As with some of the other locations, Jurien 25m doesn't appear much affected at the community level, but has seen strong changes since 2012-2013. Perhaps warrants discussion similar to that for Rottneest.

Line 224-225 As well as the heatwave impact, the 15m sites at Rottneest unlike the 15m at the other two locations has not looked like returning to pre-heatwave communities. I suggest this is something worth noting and perhaps worth some thought for the reasons in the discussion.

Line 230 Changes being driven by fine branching algae and sponges really only appears to be at 40m when looking at the figures, with minor fluctuations at 25m, but nothing appears significant. Suggest rewording the text to reflect this.

Lines 233-236 This sentence is not entirely accurate with sponges exhibiting no effect of year or year x depth at Jurien and there was no interaction term for sand at Rottneest. Suggest clarifying the text.

Line 253 Suggest Frolicher et al 2018 as a better reference

Line 258-261 Here I believe you can discuss that these species have still not recovered in the shallows and the role of deeper sites in seeding (or not) shallower impacted sites.

Lines 270-272 I am not convinced that Ecklonia is you best example here. Fig 4a seems to indicate that there was little effect on Ecklonia % cover at any sites or any depth. Indeed at

Jurien the figure suggest those at deeper sites did worse off. If patterns are hidden because of the scale of the figures then I suggest the authors do a better job of explaining this pattern in the results section.

Line 272-272 The *Scytothalia* suggests this might not be the case (yet?) and I suggest this needs acknowledging with some discussion on the potential barriers for why recovery may not have occurred yet.

Line 275 Suggest Frolicher et al 2018 as a better reference

Line 326 Delete 'only in the' and replace with 'at'. Delete reference to Figure 3e

Figure 3 legend Add 'at' after structure. Also it would seem more sensible to have the graphs ordered based on their latitudinal position and the order they are discussed in the text. I would therefore move the figures for Rottneest below those of Jurien

Fig 4a It is unclear whether *Ecklonia* totally disappeared from 15m and 40m site or if it is simply being covered by the 25m symbol. If it is the later I suggest jittering the symbols. If the former I suggest making clear in the results text.

Fig 5 Suggest mentioning in axis legends the unit of temperature measurement

Table S1 Suggest making it clear in the table legend what ND stands for. Also 2011 Coral Patches 40m is blank. Should it be ND?

Appendix B

Response to Reviewers comments on Manuscript ID RSPB-2019-0894

We thank the reviewers for their thoughtful comments that have contributed substantially to this manuscript. In particular, we attempted to specify that the lowered response of deep benthic communities to a marine heatwave shows that they acted as refuge, and that this could be evidence for their role as refugia if there was a similar response to future disturbances. We have also discussed other processes that are important to take into consideration to fully support the refugia argument. Additionally, we discuss potential mechanisms for the lowered response to the marine heatwave, despite the fact that temperature averages and anomalies were similar across all depths. Below, we provide detailed responses to each comment and their location in the tracked-changes manuscript and in the clean manuscript.

Response to Referee 1

No.	Comment / Revision	Author response	Revised (track changes) manuscript location	Revised (clean) manuscript location
1	Line 22 suggest adding 'after the heatwave' following recovery	"after the heatwave" has been added	22	22

2	Line 35 Delete refugia as glacial refugia are not just found for forest species, but a range of species in terrestrial and marine habitats.	Our aim with this sentence was not to state that refugia only occurs in forest species. Our intention was to highlight the fact that our current understanding of refugia is based on the Quaternary phylogeographic studies of biota in North America and Europe during periods of significant temperature changes. This results in knowledge gaps to predict refugia under a more contemporary theory that incorporates anthropogenic climate change. We have rephrased this sentence for clarity.	35-37	35-37
3	Line 44 Suggest Smale et al 2019 is a better reference here as it focuses on the biological responses rather than how marine heatwaves have changed	This reference has been changed as suggested	45	44
4	Line 55 Frolicher et al 2018 Nature 560: 360-364 is a better reference about future MHWs as this ms uses climate models to predict the future whereas Oliver is more focused over the past 100 years.	This reference has been changed as suggested	58	55
5	Line 64 Suggest being a little more specific about the mechanism when suggesting impacts	We have made the suggested changes and rephrased to “depends on processes such as	67-69	64-66

	will depend on location and species affected. I would suggest including factors such as position in thermal range, potential for local thermal adaptation, prevalent current, dispersal ability etc.	population connectivity, fluctuations in fecundity, post-settlement success and altered species interactions”.		
6	Line 70-79 Given that the paper focuses a lot on canopy forming algae I suggest the authors refer to some of the impacts to e.g. kelp communities. Perhaps also worth mentioning tropicalisation of marine fauna leading to altered herbivory.	We have made the suggested changes and added: “Kelp beds were lost across ~2300 km2 causing a 100 km range contraction. Kelps and other macroalgae were replaced by less complex turf-forming algae and the recovery of kelp suppressed due to the grazing pressure driven by an increase in tropical herbivory fishes”	76-79	73-76
7	Line 85 Suggest you can delete ‘effect of’	We have deleted this whole sentence to avoid mentioning at this stage other reasons why deeper habitats may be buffered from disturbances.	93-94	79
8	Line 119 Add ‘across’ after transects	We have added the word ‘across’ after ‘transects’	131	120
9	Line 172 Should the first mention of site in this line actually be location?	This is correct, we have changed the word ‘site’ for ‘location’	185	174
10	Line 178-188 I found this section a little confusing to read and also some important	Thank you for pointing this out. We have rephrased and corrected these paragraphs,	183– 203	170-180

	information was missing. It seems to me that the two paragraphs can be combined as they appear to be explaining the same type of analysis. The confusing aspect relate to the fact the data seems to have been treated differently e.g. some used Euclidean distance (appropriate for univariate PERMANOVA) and some used Bray-Curtis resemblance matrix. Some clarity here would be helpful. Also the authors need to state with model they used to analyse the data (this would be different for the multivariate and univariate analyses) and also how many permutations were undertaken to determine significance. The authors also need to state why a dummy variable was added.	which also contained errors. We deleted lines 178-180 since it was repeating the paragraph bellow as you pointed out. In line 187, ‘Bray-Curtis similarity’ was replaced by ‘Euclidian distance’. To clarify, we treated data differently: for the multivariate tests we used Bray-Curtis similarity which is more appropriate for community composition comparisons; for the univariate PERMANOVA analyses were treated the data with Euclidean distance because it is analogous to traditional ANOVA.		
11	Line 195 I think the authors need to make sure the reader is clear they are discussing shallow waters here as Fig 2b does not suggest this assemblage for deeper waters. Also suggest being specific here and sating Sargassum to	Thank you for this comment. We have reworded the sentence to make it clear that we are discussing shallow assemblages and	211-216	195-199

	avoid confusion with other large macroalgae that are discussed further into the paragraph.			
12	Line 198 Fig 2b doesn't indicate that Ecklonia is important. I am that it was present so perhaps worth stating something about drivers of community structure.	Ecklonia was present in shallow sites at Abrolhos, but as noted, it was not highlighted in the PCO as a driver for that community, so we have deleted ' E. radiata ' from line 198.	215	198
13	Line 199 I don't agree that fine-branching red algae seems to be a driver for deeper Abrolhos sites and Jurien Bay. The only driver seems to be sand. Reference to percent cover sand should be Fig S5.	We agree, and have deleted 'fine branching red macroalgae'. We have examined and changed the supplementary material and fixed the order and name of the figures in the manuscript accordingly.	215-216	198-199
14	Line 204 See comment above	We have deleted 'fine branching red macroalgae'.	220	204
15	Line 208 Fig S2 doesn't show anything about bleached coral. It states how the percent cover of coral changes. Suggest rewording.	We have examined and changed the supplementary material, we have included a figure illustrating bleached and unbleached coral at Abrolhos Islands and fixed the order and name of the figures in the manuscript accordingly.	227	208
16	Line 209 Throughout the SOM the figures are not in the order stated in the text. Please make sure this is corrected.	Thank you for this comment. We have examined and changed the supplementary	Throughout manuscript	Throughout manuscript

		material and fixed the order and name of the figures in the manuscript accordingly.		
17	Line 210 Reference to foliose coral should be S2. Also make clear that you are referring to shallow benthic communities in the next sentence.	We have examined the supplementary material and fixed the order and name of the figures in the supplementary material and manuscript accordingly. We have made clear that we are referring to shallow communities in the next sentence by adding ‘shallow (15 m)’ after ‘benthic community’.	228-229	209-210
18	Line 211-214 Suggest the authors make it clear that it appears that these species have not recovered since the heatwave. Also S5 should be S3 and S6 should be S4.	We have added a sentence indicating that 25 m communities at Abrolhos Islands have not returned to pre-heatwave conditions. We have examined the supplementary material and fixed the order and name of the figures in the supplementary material and manuscript accordingly.	233-234	214-215
19	Line 219 Add ‘at 15m’ after recovery	We have added ‘at 15 m’ after recovery	239	220
20	Line 220 S6 should be S4	We have examined the supplementary material and fixed the order and name of the figures in the supplementary material and manuscript accordingly.	240	221
21	Lines 216-222 As with some of the other locations, Jurien 25m doesn’t appear much	Thank you for this comment. We have added a sentence indicating ‘Communities	239-244	220-225

	affected at the community level, but has seen strong changes since 2012-2013. Perhaps warrants discussion similar to that for Rottnest.	at 25 m seemed to be affected by a separate event to the 2011 heatwave, since community composition was recovered to pre-heatwave conditions in 2012 and by 2013 it show change towards a more turf driven community’.		
22	Line 224-225 As well as the heatwave impact, the 15m sites at Rottnest unlike the 15m at the other two locations has not looked like returning to pre-heatwave communities. I suggest this is something worth noting and perhaps worth some thought for the reasons in the discussion.	Thank you for this comment. We have added the sentence ‘In contrast to the other two locations, community composition at shallow sites in Rottnest Island had not recovered to pre-heatwave conditions by 2017’.	250-251	230-232
23	Line 230 Changes being driven by fine branching algae and sponges really only appears to be at 40m when looking at the figures, with minor fluctuations at 25m, but nothing appears significant. Suggest rewording the text to reflect this.	We have added ‘at 40 m sites’ and rephrased the sentence.	255-256	236-237
24	Lines 233-236 This sentence is not entirely accurate with sponges exhibiting no effect of year or year x depth at Jurien and there was	We have added a sentence to make this statement more accurate which states ‘Exceptions where sponges at Jurien Bay, which did not exhibit an effect of year or its	261-263	242-244

	no interaction term for sand at Rottnest. Suggest clarifying the text.	interaction with depth, and sand at Rottnest Island which showed no effect with the interaction of year and depth'		
25	Line 253 Suggest Frolicher et al 2018 as a better reference	We have included Frolicher et al 2018 as a reference in this section.	280	261
26	Line 258-261 Here I believe you can discuss that these species have still not recovered in the shallows and the role of deeper sites in seeding (or not) shallower impacted sites.	We value your comment about the seeding sources and how their existence should result in the recovery of population they are acting as refugia for. Our aim with this study is a step a step before that, because for a population to be refugia, it first needs to be somehow buffered, protected or resistant to biophysical disturbances. If a population or community shows such effect, then it can potentially be a source of propagules. Nonetheless, the fate of these propagules will depend on other processes not evaluates in this study, such as dispersal potential, current direction, fecundity and post-settlement survival. These processes are beyond the scope of this manuscript, and our attempt is to highlight how the persistence of this communities at depth is the first step to a potential refuge from	287-288, 300-301, 361-368	268-269, 281-282, 341-348

		extreme events. We have incorporated your comment in the discussion in this particular section, by stating the main focus of the study, rather than the refugia hypothesis: ‘supporting our hypothesis of potential depth buffering effect from extreme climatic events that allows the persistence of kelp dominated communities’. We have further incorporated your comment in another section of the discussion, since we believe is extremely important to evaluate other aspects of the deep-refugia hypothesis (as listed above), as well as the further study of general processes driving off-shore communities.		
27	Lines 270-272 I am not convinced that Ecklonia is you best example here. Fig 4a seems to indicate that there was little effect on Ecklonia % cover at any sites or any depth. Indeed at Jurien the figure suggest those at deeper sites did worse off. If patterns are hidden because of the scale of the figures then	Our aim with this sentence is to show that key habitat-forming species are present at all our sites, and specifically that the deeper sites did not show the same magnitude of reduction as shallow sites. Although, in some cases deeper sites show a small response as you highlight, in the case of 15 m Jurien Bay, a response the shallow	298-300	279-281

	I suggest the authors do a better job of explaining this pattern in the results section.	communities was not noticeable because the Ecklonia cover at this sites is marginal due to the dominance of sandy substrate. This sentence has been reworded to make it more accurate: ‘key habitat forming species like Ecklonia , Scytothalia and fine branching red algae (figures 5a, 5b, and 5c) were found along the transition zone and across all depths, and showed minimal or no reductions in deep sites’		
28	Line 272-272 The Scytothalia suggests this might not be the case (yet?) and I suggest this needs acknowledging with some discussion on the potential barriers for why recovery may not have occurred yet.	We value this comment and have added the sentence: ‘although, Scytothalia in shallow communities had not recovered in 2017’.	300-301, 361-366	281-282, 341-354
29	Line 275 Suggest Frolicher et al 2018 as a better reference	We have included Frolicher et al 2018 as a reference in this section.	305	285
30	Line 326 Delete ‘only in the’ and replace with ‘at’. Delete reference to Figure 3e	We have replaced ‘only in the’ with ‘at’ and deleted the reference to figure 3e.	360	340
31	Figure 3 legend Add ‘at’ after structure. Also it would seem more sensible to have the graphs ordered based on their latitudinal position and the order they are discussed in	We have added the word ‘at’ after structure. We have also realised that the Rottnest Island and Jurien Bay PCO’s were labelled incorrectly, so that the graphs were in	602-605	581-584

	the text. I would therefore move the figures for Rottneest below those of Jurien	latitudinal order, but the legend was not. This has been fixed and the figure and legends are now in latitudinal order.		
32	Fig 4a It is unclear whether Ecklonia totally disappeared from 15m and 40m site or if it is simply being covered by the 25m symbol. If it is the later I suggest jittering the symbols. If the former I suggest making clear in the results text.	The 15 m sites where covered by the 25 m sites and symbols have been jittered for clarity. The 40 m sites at Abrolhos Islands were not surveyed in 2011 due to technical problems (electronic supplementary material, Table S1)	664	636
33	Fig 5 Suggest mentioning in axis legends the unit of temperature measurement Table S1 Suggest making it clear in the table legend what ND stands for. Also 2011 Coral Patches 40m is blank. Should it be ND?	We have added '(in degrees Celsius)' in the legend of figure 5. We have also added what ND stands for in supplementary table S1: 'ND = No Data, sites not surveyed that year', and added the missing ND to 2011 40 m Coral patches.	620, electronic supplementary material	598, electronic supplementary material

Response to Referee 2

No.	Comment / Revision	Author response	Revised (track changes) manuscript location	Revised (clean) manuscript location

1	Line 51-53: Is this more specific to all refugia or refugia in BTZ's? The set up would seem like BTZ's, but it seems like a more general statement. The flow seems to be about BTZ's and it comes off a bit awkward that they would be the justification for identifying refugia.	Thank you for this comment. The statement is general, rather than specifically about BTZ's. Our reference to BTZ's is to describe our study area and highlight that they are hotspots for biodiversity and also very vulnerable to extreme climatic events. We have changed the order of this paragraph for clarity.	44-56	43-53
2	Para beginning Line 54: If marine heat waves are also involved in the process of coral bleaching, which the originators of the concept state (http://www.marineheatwaves.org/all-about-mhws.html), and this is one of the most pressing biodiversity crises globally, seems like it deserves a mention in this paragraph.	This deserves mention, so we have added coral bleaching to the list of devastating effects associated with marine heatwaves.	61-62	58-59
3	Line 74: "the bleached coral was reported to be 12-100 %". What do you mean specifically? The individual coral colonies has 12-100% of living tissue stark white bleached? Please be more specific.	Thank you for your comment, this really should be more specific. We have revised the reference (Abdo et al, 2012) and changed the numbers accordingly. In the reference, they categorised the bleached coral in different categories which we have not included for brevity.	81-82	77-78

4	Line 85-86: I would avoid bringing up other reasons that deeper habitats might be buffered, such as local anthro pollution, as it just detracts from the point you are making about depth refugia from thermal stress and ecological change.	Thank you for this suggestion. We have deleted this part from the discussion.	93-94	86
5	Line 83: I would say “may benefit” because it is not necessarily true in all systems across all events. E.g., Smith TB, Gyory J, Brandt ME, Miller WJ, Jossart J, Nemeth RS (2016) Caribbean mesophotic coral ecosystems are unlikely climate change refugia. Global Change Biology 22:2756–2765	We have added the word ‘may’ before ‘benefit’	91	84
6	Lines 101-105: First, this is a long sentence. Second, you cannot argue that deep benthic habitats (as in all habitats) are a refugia based on a single study (one event) from Western Australia? Rephrase this to make it more specific to the habitat where your study was conducted. Also, a refugium can only be determined over multiple events. A refuge can be determined for one event. So, you are going to have hard task making a convincing argument that	Thank you for this criticism. Incorporating into this manuscript makes it much more accurate description of what we are proposing. We have rephrased the sentence to make it specific to our study area. We have also changed the word refugium to refuge as per your suggestion, since it is true we are only evaluating one disturbance event. Nonetheless, we have kept the idea, that if deep reefs were	110-116	101-105

	this represents a refugium without more evidence over more events.	buffered from future extreme events, then these could act as refugia.		
7	Line 119: Change to “transects of the seafloor”	We have added the word ‘across’ as suggested by referee 1.	131	119
8	The paragraphs starting on line 205 that introduce the site changes over time have numerous errors in reference to the supplementary material. Below are a couple of examples, but nearly all references to the supplemental figures are incorrect.	We have revised all the references to supplementary material and fixed the numerous mistakes in the manuscript and in the supplementary material. All figures and tables are now referenced correctly.	Throughout text and in electronic supplementary material	Throughout text and in electronic supplementary material
9	Line 208: The Fig. S2 presents no information on coral bleaching. Also, the figure only lists Foliose coral cover. Is this the only morpho-type that was present? What about the massive, encrusting, and staghorn/tabulate presented in Fig. 2?	Thank you for this comment, we have realised that there were several errors in the way the figures were presented in the supplementary material. These have all been fixed in the supplementary material and in the text. Figure S2 now illustrates bleached and unbleached coral and figure S4 on foliose coral at Abrolhos Islands. There were also staghorn, tabulate, massive and encrusting coral morpho-types at Abrolhos Islands. These had not	226, Electronic supplementary material	

		been included in the supplementary material, but we have included graphs staghorn and tabulate coral as figures S8, S9 respectively. Individual figures for massive and encrusting coral are not included as they account for less than 1% of coral cover.		
10	For both coral and turf you use the term “significant” implying a statistical test, but I do not see an associated statistical test aside from the general PERMANOVA. The PERMANOVA is showing the overall depth*time trend individually at each location. It is not testing specifically for an increase or decrease in any time period, which is what your statement says is “significant”. The use the term “trend” in introductory sentence (Line 205), so perhaps this is not statistical, in which case the term “significant” should be avoided. Alternatively, you could probably construct tests of change over time to support you contention of an increase (e.g., a significant increase in turf at Abrolhos after the 2011 marine heatwave).	Thank you for this comment. This sentence wanted to describe a substantial increase in bleached coral and turf, since as you note, our general PERMANOVA does not test for increase or decrease over a specific time period. We have changed the word ‘significant’ for the word ‘substantial’ in the manuscript for accuracy in the statement.	227	207

11	For sand, how does including an abiotic variable that should not respond to a marine heatwave affect the rest of the data in the PCO?	We included the percent cover of substrate (sand or hard substrate) as a proxy for an increase or decrease in general biotic cover and available space. Our classification of the benthic biota, although general (only key species were identified, the others were classified by functional morphology), includes an extensive number of taxa and morphologies. As such, if there was a general decrease (or increase) in several functional morphologies that was not substantial enough per class, but enough as group to be noted in the multivariate analysis, this would be captured as an increase (or decrease) in the availability of substrate (namely sand or hard substrate).		
12	Figure 4c. Is the data point for Abrolhos 40m hidden in 2011 or not there? You should not in the caption.	Thank you for this comment, we have jittered the symbols to ensure all the sites available are visible. This has improved the graphs considerably.	664	637

13	Line 225-226: “At the 15 m sites there was an increase in E. radiata (figure 4a)”. This doesn’t look like much of a change to me. It is especially unconvincing without a statistical comparison.	Thank you for this comment. This is true, and the increase of E. radiata at 15 m sites is slight. We have deleted this part of the sentence: ‘an increase in E. radiata (figure 4a), but’ for statement accuracy.	249	224
14	Line 237: Use past tense “ranged”	The word ‘ranges has been replaced for ‘ranged’	263	244
15	Figure 5. In cases where there is no data point visible (either behind another data point or there was no data) please indicate in the caption if there is a data point and where it is. E.g., Abrohos 15 and 25 m in 1995 are overlapping as indicated by Table S3, but you would have to dig into the ESM to figure that out.	Thank you for this comment, we have jittered the symbols to ensure all the sites available are visible. This has improved the graphs considerably.	671	
16	Figure 5b. A 0 line would be helpful for reference.	A line 0 has been added to figure 5b as suggested	666	644
17	Inconsistency in the format of literature citations in the References section. There also seem to be notes to finalize references (e.g., #45 “double check”)	Working on this while you get back to me on the other comments	393-587	371-565

Appendix C

The paper by Giraldo-Ospina et al deals with a very timely subject: the impacts of marine heat waves (MHW) in marine ecosystems. MHWs are documented in many areas around the globe and their intensity and frequency are expected to increase under climate change. Thus understanding the impacts of MHW is one of the pillars to address management options to sustain resilient ecosystems and the key ecosystem services that they support.

The paper explores how a MHW recorded in W Australia impacted the benthic communities across a subtropical-temperate biogeographical transition zone and across depth from 15 to 40m. Most scientific literature on MHWs focused in shallow habitats (from 0 to 10-15m), much less studies characterized the impacts in deeper communities. Thus, this study is filling the gap of knowledge. The results indicate that deep reefs have the potential to act as refugia against MHWs. This is an important result since deep reefs may play a role in the recovery of foundation species wiped out from shallower areas.

I think that the paper can be accepted with minor revisions.

The authors should address the following points:

Material & Methods

Lines 126-129 -Indicate the area covered by the each of the 30 images analyzed.

Lines 126-129 -Please justify why you decided to use 30 not overlapping images.

In my opinion using overlapping images this is one of the strength of monitoring permanent plots in order to be able to quantify community changes. I understand that the reason behind is the statistical issues of non-independent data over time. I'm not specialist in statistical analysis but I think that there are methods to deal with this point. Of course, I am not asking to the authors to redo the image analysis! But I think that they should take this comment in consideration in future studies. Specially because, in my opinion, at some extent some of the changes observed through time in the different locations and depths could be, at least partially, linked to spatial heterogeneity in the benthic communities (see below).

Lines 158-164-Temperature data. The authors should provide information on the number of data points included per year/ depth and location in the analysis. I'm sure that the Australian Shelf Temperature Data Atlas surely collected already indicated these data. Since the paper is dealing with the effects of spikes of temperature and not means trends this information is key for the interpretation of the results. Probably the data missing in Supplementary table S3 is linked to the amount of data availability to be considered.

Lines 158-164-Why the temperature a 30 km radius from the study locations for the analysis was chosen? In 30 km temperature conditions can show relevant differences for organisms.

Lines 169-176-Please justify why for the characterization of communities the centroids in the PCO analysis represent averages for each site per location, depth and year resulting from 2 or 3 grids while in the analysis of the trajectories represent the averages for each depth per year derived from 4 to 6 grids.

Lines 169-176-Please justify why in Fig 3 the differential thickness of the arrows. Arrow lengths seems not to be linked to the thickness.

Results

-It would be great to include some pictures showing before and after 2011 MHW changes / no changes

-The sites should be represented by the same symbols across all figures. This will help the readers.

-Line 197 something is missing before the reference to the figures.

-I'm not familiar with the W Australian communities but having "Sand" and "Hard Substrate" as group characterizing some of the benthic communities (see Fig 2 and 3) is not straightforward.

-What species of seagrasses were found in the surveys? Are these species developing with corals, kelps and Sargassum?

-In Figure 3, some transitions after the 2011 have similar vector lengths as the ones from 2010 to 2011 which are attributed to the 2011 MHW.

Discussion

-In some cases, the community composition was different across depth before 2011. For instance, in Jurien Bay Shallow was characterized by sand and seagrasses at 15 and 25 but not at 40 m sites (Supplementary figure S6). These differences should be taken into consideration in the discussion of the potential role of deep communities as refugia. The authors are not comparing the same communities.

-In the same line, to what extent the observed patterns are due to the sampling strategy (comparing non overlapping sampling areas) should deserve some attention in the discussion. In my view, comparing the same areas should provide much more robust patterns and conclusions. But the authors know the sampled areas and maybe this consideration make no sense in this case.

-How many species can be included in the "Fine branching red algae", "Coarse branching red algae", "Encrusting red algae", "Encrusting brown algae" and other functional/morphological groups? In the literature there are studies showing that similar morphology not always matches with functions and responses to stressors. I think this point deserves some consideration in the discussion of the deep communities as refugia.

Appendix D

Depth moderate loss of marine foundation species after an extreme marine heatwave: Could deep temperate reefs act as refuge?

Response to Reviewers comments on Manuscript ID RSPB-2019-2625

We thank the reviewers for their comments that have substantially improved this manuscript. Specifically, we have explained the use of terminology defining the difference between refuge and refugium. This, with the aim to clarify that the lowered response of deep benthic communities to a marine heatwave shows that they acted as refuges, and that this is evidence for potential refugium (not proof of it). Their potential role as refugia has been addressed in the discussion only. Additionally, and as requested by the editor and reviewers, we have changed one of the figures (figure 4) to allow for a more clear comparison of post-heatwave changes in percent cover on important macroalgae species, compared to what had been previously reported for shallow inshore reefs. Below, we provide detailed responses to each comment and their location in the tracked-changes manuscript and in the clean manuscript.

Response to Referee 3

No.	Comment / Revision	Author response
1	Lines 126-129 -Indicate the area covered by the each of the 30 images analyzed.	We have added this information: ~ 4m ² each image
2	Lines 126-129 -Please justify why you decided to use 30 not overlapping images.	In preparation for this study, we run a power analysis to calculate the number of images needed to representatively sample the benthic communities while optimizing resources (time). This analysis was important, since the classification of each image requires between 4-8 minutes and we had a total of 202 grids to process with over 1000 images each.
3	In my opinion using overlapping images this is one of the strength of monitoring permanent plots in order to be able to	Thank you for this comment.

	quantify community changes. I understand that the reason behind is the statistical issues of non-independent data over time. I'm not specialist in statistical analysis but I think that there are methods to deal with this point. Of course, I am not asking to the authors to redo the image analysis! But I think that they should take this comment in consideration in future studies. Specially because, in my opinion, at some extent some of the changes observed through time in the different locations and depths could be, at least partially, linked to spatial heterogeneity in the benthic communities (see below).	For this study, we decided to use a PERMANOVA when comparing the percent cover of individual benthic categories because is a robust method, especially when your design is heavily unbalanced. We are looking at other ways to examine this type of data for future studies and in the case of these sites, since they continue to be part of a long-term monitoring plan.
4	Lines 158-164-Temperature data. The authors should provide information on the number of data points included per year/ depth and location in the analysis. I'm sure that the Australian Shelf Temperature Data Atlas surely collected already indicated these data. Since the paper is dealing with the effects of spikes of temperature and not means trends this information is key for the interpretation of the results. Probably the data missing in Supplementary table S3 is linked to the	We have added the details of the temperature data requested in as a table (table S5) in supplementary material. The temperature data that we were able to obtain was used to create plots, since the data for some locations was very scarce. There were years, were only one data point exists for Abrolhos and Jurien Bay sites (no error bars in the plot), and there are years where there is no summer temperature data at all depths as can be seen in the temperature data. We acknowledge the weakness of our temperature data inference, yet the data used is the only data that exists. This information gap needs to be addressed in order to fully understand the mechanisms that drive offshore communities.

Depth moderate loss of marine foundation species after an extreme marine heatwave: Could deep temperate reefs act as refuge?

	amount of data availability to be considered	
5	Lines 158-164-Why the temperature a 30 km radius from the study locations for the analysis was chosen? In 30 km temperature conditions can show relevant differences for organisms.	The 30 km radius was an arbitrary number chosen on the basis of getting as much temperature data as possible for each location at depth within a considerable range. We were very limited by the temperature data for this study, since there are not thermistors located at each depth of our study sites. That is why the data obtained was from “vessels of opportunity”. That means, that any temperature measurement that has been collected under the IMOS facility in Australia would be in this data set. Including gliders, AUVs, CTD casts and any other. These data is far from ideal, but is all we have to compare to benthic data and it does give an idea of temperature variation, with respect to depth.
6	Lines 169-176-Please justify why for the characterization of communities the centroids in the PCO analysis represent averages for each site per location, depth and year resulting from 2 or 3 grids while in the analysis of the trajectories represent the averages for each depth per year derived from 4 to 6 grids.	The community composition across the whole transition zone (three locations, figure 2) was based on the average of the 2 or three grids collected for each site (there were two sites, per depth and location, figure 1, except in Abrolhos Islands, where there were three sites for the 25 m depths). For the multivariate analysis for each location (figure 3), we averaged all the grids from both sites (6-4 grids). This would allow us to track the trajectory of the overall community composition for each depth and each location.
7	Lines 169-176-Please justify why in Fig 3 the differential thickness of the arrows. Arrow lengths seems not to be linked to the thickness.	In figure 3, the thicker arrows are the ones that point to the change in community composition from 2010 to 2011. We did this to highlight the response to the MHW. (Lines 615-617, figure legend)
8	It would be great to include some pictures showing before and after 2011 MHW changes / no changes	We could include some of these pictures, though this would only be obvious for overlapping images. Since we subsampled the total number of images obtained per grid (over 1000 images were captured per grid) it is unlikely that we measured the same point across all years.
9	The sites should be represented by the same symbols across all figures. This will help the readers.	Sites are represented by the same symbols in figures 2 and 3. The sites are all squares in the map (figure 1) as they represent the square grids that were surveyed by the AUV (not to scale).

Depth moderate loss of marine foundation species after an extreme marine heatwave: Could deep temperate reefs act as refuge?

10	I'm not familiar with the W Australian communities but having "Sand" and "Hard Substrate" as group characterizing some of the benthic communities (see Fig 2 and 3) is not straightforward.	We have included these categories as they describe the availability of substrate, which may describe that communities are transient or only sub-sediment (for sand) and that there is available substrate for colonization (for rock or hard bottom). Some of the deep sites (like Jurien), presented seagrass and sand and were very variable across years.
11	What species of seagrasses were found in the surveys? Are these species developing with corals, kelps and Sargassum?	We were not always able to identify the species of seagrass, but when possible these were identifiable to genus: Amphibolis sp. and Posidonia sp. Sometimes, these were present in a mixed substrate of sand and rock, in which case they were living with other macroalgae and sponges. Since corals were mostly found in the shallow sites of Abrolhos islands, we did not identify seagrass and coral together in our sites.
12	In Figure 3, some transitions after the 2011 have similar vector lengths as the ones from 2010 to 2011 which are attributed to the 2011 MHW	The 2010-2011 vectors are the ones attributed to the MHW. Despite these surveys occurring in April, so when the MHW had not yet abided (in 2011), the temperature anomalies that already been occurring for months. The change in community composition is visible in the PCO plots. The 2012 surveys, were carried out a year later, were shallow communities (most impacted by the HW) were returning to pre-heatwave composition.
13	In some cases, the community composition was different across depth before 2011. For instance, in Jurien Bay Shallow was characterized by sand and seagrasses at 15 and 25 but not at 40 m sites (Supplementary figure S6). These differences should be taken into consideration in the discussion of the potential role of deep communities as refugia. The authors are not comparing the same communities	It is true that the community composition varies across depth naturally. We address this specific point in figure 2, where we aim to provide a big picture summary of the different assemblage groups that occur at all locations across the biogeographic transition zone.
14	In the same line, to what extent the observed patterns are due to the	Thank you for pointing this out. The observed patterns we believe are very robust, in the sense that the areas surveys each time were the same. Of course, not all the

Depth moderate loss of marine foundation species after an extreme marine heatwave: Could deep temperate reefs act as refuge?

	sampling strategy (comparing non overlapping sampling areas) should deserve some attention in the discussion. In my view, comparing the same areas should provide much more robust patterns and conclusions. But the authors know the sampled areas and maybe this consideration make no sense in this case	grid was measured (as pointed out previously, this is not feasible due to the time it takes to classify images). But we made our best, to ensure representative subsampling of images to ensure a robust analysis. Moreover, the sampling strategy was able to detect regions of natural high variability. For example, Jurien Bay is a location where we found high natural variability in community composition. This finding reinforces the importance of long-term monitoring of deep-offshore locations, which is key to allow us to detect changes that are outside the natural ranges.
15	How many species can be included in the “Fine branching red algae”, “Coarse branching red algae”, “Encrusting red algae”, “Encrusting brown algae” and other functional/morphological groups? In the literature there are studies showing that similar morphology not always matches with functions and responses to stressors. I think this point deserves some consideration in the discussion of the deep communities as refugia	It is impossible to say for this specific study how many species were pooled into each of the functional/morphological categories we used. What you point out is an important consideration, which unfortunately we don't have the space (word limit) to include in the discussion. The main outcome of this study is that deeper communities responded less to a marine heatwave and that offshore habitats across all depths responded less than inshore ones (what has been reported in the literature). Ultimately, the role of refugia for each functional group or foundation species (E. radiata or S. dorycarpa) would need to be considered individually in detail and by answering other questions: are they reproductive at depth? Is there oceanographic connectivity between deep and shallow reefs? Overall, this study gives a first insight and opens the door to study cross-depth patterns more in detail to either enforce or refute the potential for refugia.

Response to Referee 4

No.	Comment / Revision	Author response
1	Suggest dropping the second clause from the title as the paper does not answer the question of whether deep reefs act as climate refugia	Thank you for this suggestion, we have not dropped the entire clause, but we have changed the word refugia for refuge.

Depth moderate loss of marine foundation species after an extreme marine heatwave: Could deep temperate reefs act as refuge?

2	Lines 47 and 92. Might want to cite Ladah and Zertuche-Gonzalez 2007 (Botanica Marina 47: 367–372) since they argued deep reefs served as a refuge for the giant kelp in Mexico during the 1997-98 El Nino.	Thank you for this suggestion, we have added this reference.
3	Lines 158-167. Might consider looking at metrics other than the deviation from the mean (e.g. maximum, or 90th percentile) given that many species exhibit a threshold response to temperature stress	Due to the nature of the temperature data set we used (vessels of opportunity), there were years for which we were only able to obtain two or one temperature measurements for each depth. This restricted the temperature comparisons we were able to make, and hence we decided to use a comparison with the mean temperature.
4	Line 212 Fig S5. Confusing statement as the increase in Sargassum in Fig. S5 was relatively small at 25m (as was Ecklonia), while the increase in Sargassum cover was comparatively higher at 15 m. Need to clarify	The Sargassum figure has been rearranged and it is now in figure S7 in the supplementary material. This has been changed throughout the manuscript. We have reviewed the figures and the original statement is correct, and the increase of Sargassum at 25 m (yellow circles) in Abrolhos is visible, with no increase at 15 m (red circles). A similar pattern was observed for E. radiata at this site and depth (figure S6).
5	Line 220. Figure 4c shows encrusting red algae. Should the reference be Figure 4d which shows fine branching red algae? Regardless, there is little fine branching red algae at any depth in any year at Rottneest according to Fig. 4, so it's hard to make much of this statement	We have changed figure 4 entirely to focus on one of the main findings of this study: Offshore reefs at all depths did not show a response to the MHW at the scale reported for inshore reefs. Figure 4 now depicts 4 macroalgae (E. radiata , turf, encrusting red and S. dorycarpa) as these were reported for inshore reefs, and compares our findings to those of Wernberg et al., (2013)
6	Line 230-131. The emphasis on recovery in community composition here seems a bit misleading given that Figure 4 shows that the % cover of Ecklonia and encrusting red algae (which formed the majority of the cover at shallow depths at Rottneest) remained relatively constant during the time series. Consider revising to more	We agree with what you point out here. However, this specific sentence refers to the multivariate analysis rather than individual species change in percent cover (figure 3d). The community composition of shallow sites (15 m) for Abrolhos (figure 3a) and Jurien (figure 3c), had returned to levels similar to pre-heatwave composition. However, this pattern is not seen in the shallow reefs of Rottneest. Despite E. radiata not changing much at this location, S. dorycarpa decreased and Sargassum sp. increased (in shallow sites).

Depth moderate loss of marine foundation species after an extreme marine heatwave: Could deep temperate reefs act as refuge?

	accurately describe the changes and/or lack of change observed	
7	Lines 235-236. Need to provide a more detailed description of the magnitude of these changes (e.g. sponges increased from ~ 4% in 2010 to ~ 9% in 2017).	We have added this information (Lines 251-254)
8	Lines 236-238. Again, it does not seem sufficient to simply state that S. dorycarpa declined. The reader needs to know how much it declined. In this case it looks like average cover of S. dorycarpa varied between about 3 and 7% at 40 m during the time series. The details matter because you are arguing for importance of deep reefs in providing refuge during a heatwave and in serving as refugia for the recovery populations in shallower areas that were decimated by the heatwave	We have added this information (Lines 251-254)
9	Line 2560-262. need to be more clear about what the evidence is that you are referring to. The specific taxa need to be identified	We have specified the taxa we were referring to: E. radiata and S. dorycarpa (Line 291)
10	Lines 265-269. The purported depth buffering is not evident from this study. Catastrophic loss was only shown for S. dorycarpa and this species was largely absent or rare at the two northern sites at all depths. In the case of Ecklonia there seemed to be little difference in temporal response to heatwave among the three depths at Rottneest where it was most abundant. It was never present shallow at Jurien, or deep at Abrolhos	The effect of depth is evident in regards to community composition (figure 3), were community composition in deeper sites changed less following the 2011 marine heatwave. In regards to the foundation species (E. radiata and S. dorycarpa), they were less affected than their counterparts in inshore reefs (figure 4). The case for S. dorycarpa only applies to Rottneest (since it was not or rarely present at Jurien and Abrolhos), yet their percent cover was less reduced at deeper sites. This study provides a first insight into what are the dynamics of deeper communities, and how they respond to extreme disturbances. From this point, further assessments are required to prove the existence of deep refuge and to elucidate the mechanisms that lead to it.

Depth moderate loss of marine foundation species after an extreme marine heatwave: Could deep temperate reefs act as refuge?

11	Lines 273-277. This is another instance of where information on the magnitude of change as well as the direction is needed to provide context for evaluating ecological significance.	The section referred to here, is a summary statement at the beginning of the discussion. Details on the magnitude of changes have specified in the results as requested (Lines 299-304), but due to word count restrictions, this section has not been modified.
12	Lines 278-281. This statement seems a bit misleading as Scytothalia was absent or rare at all depths at Abrolhos and Jurien and fine branching red algae were rare or absent at Rottneest. Perhaps my criticism in this case is due to my confusion about what “transition zone” refers to in this case. In any case, this statement needs to be revised for clarity and accuracy	Thank you for this comment, that help us make this statement more clear. This has been re-written for clarity: “Moreover, key habitat-forming taxa like Ecklonia , and fine branching red algae, were found along the transition zone across all depths. S. dorycarpa , was only found in Rottneest Island, but showed small decreases in deep sites compared to in shallow ones.” (Lines 305-308)
13	Lines 281-285. The size of a source population as well as its fecundity and physical connectivity is important in determining its ability to serve as a refugia, which in this paper is defined as a “region that “that facilitate the temporal and spatial resilience of biological communities and enable the recovery from biophysical disturbances”. The argument that deep reef populations may be key drivers of shallow reef resilience would be much stronger if the reader was provided with some information on the sizes of the populations of interest (habitat forming species? other species?)	The referee highlights an important consideration. Currently, we are in the process of modelling the distribution of these populations with species distribution models based on a few sites that we have information for (at 25 and 40 m sites). Basic information like this is lacking for deep offshore sites because it is challenging to collect. Based on the little information we have, we can estimate populations of several km ² , maybe even 10’s of km ² , at Rottneest and Abrolhos sites. Yet, how patchy and how ‘connected’ they are, still remains to be determined. This study provides the first insight, the first clue as to whether deep habitats may be refugia by showing a reduced response in deep communities to an extreme disturbance (they acted as refuges from 2011 MHW). However, to determine the existence of refugia in these reefs, estimations of reproductive performance and oceanographic connectivity are required. Additionally, continuous monitoring, to corroborate the effect of depth from other extreme events. (Lines 311-315 in current manuscript)
14	Lines 291-292. The finding that the results of this study differed from those of Wernberg et al is quite	Thank you for this comment. One of the main findings of this study is the discrepancy with previous studies. Our study looked at different sites

	interesting given the notoriety of these earlier studies. It would be worth expanding on why the results of the two studies likely differed. Is it because different sites were sampled? Same sites but different depths? Any additional information that can be provide on this discrepancy would be useful helping to understand the nature and extent of impacts resulting from this heatwave	because we focused in offshore communities (sites across all depths were in offshore locations, deep sites of more than 25 m are only found kilometres offshore in WA), while the sites studied in Wernberg et al. (2013) were located in inshore reefs. The latitudinal extent of both studies is comparable, so the only difference is the distance from the shore of sites. This rationale for the difference between offshore and inshore responses to MHW is explained in lines 349-355 of the current manuscript.
15	Line 315. Insert “positive” before net photosynthesis	We have done so. (Line 348)
16	Line 333. These references are a bit data and the author’s might want to check this statement for accuracy. By most accounts the 2014-15 "Blob" and El Nino in the central and eastern Pacific seemed was more extreme than the 2011 heatwave in WA	We have inserted the reference of Hobday et al. (2018). They categorize marine heatwaves according to their duration and multiples of the 90 th percentile of the local climatology. According to this classification of marine heatwaves, the 2011 MHW in WA, was more extreme (category IV, Extreme) than “The Blob” (category III, Severe). The main difference between the two events is given by their duration and temperature anomaly: The Blob had a duration of 711 days, compared to 66 days of the 2011 WA MHW. However, the intensity (highest temperature anomaly value throughout the heatwave) was 2.56 °C for “The Blob” and 4.89 °C for the 2011 WA MHW. When that study was published, the only event categorised as “Extreme” (category IV), was the 2011 WA MHW. (Line 386)
17	Line 342-343. Determining the role of deep reefs as a “resilience mechanism for shallow species” not only requires documenting the response of deep reefs, but it also requires documenting the recovery of shallow reefs and the genetic or demographic connectivity evidence that shows source populations on deep reefs contributed to this recovery.	This is true, and unfortunately that is beyond our study. There are now reports of small populations of E. radiata found in areas that were completely devastated by the 2011 MHW (those reported by Wernberg et al., 2013). The source of these new populations is currently being investigated with multidisciplinary studies. Those results will help refute or add evidence for the refugia hypothesis we pose in this manuscript.

Appendix E

I believe the revised manuscript is much improved and the authors adequately addressed many (though not all) of the concerns that I raised in my review of their earlier submission. Below I list both general and specific comments for the authors to consider.

General comment

Rather than (or perhaps in addition to) the sole focus on depth as a refuge from warming one can think of the extirpation of species in shallow water following the MHW to represent a range contraction much like the range contraction at the equatorial edge. A number of climate warming studies on terrestrial plants and insects have shown that range expansions and contractions in elevation mirror those in latitude. Discussing parallels between marine and terrestrial systems in this regard might be something to consider in the Introduction and Discussion as well as the potential for refuges in depth vs refuges latitude to serve as refugia for shallow water populations extirpated by MHWs.

Specific comments

Line 31. The specific ecological catastrophe referred to in this sentence is unclear. Suggest replacing "this rapid ecological catastrophe" with "such ecological catastrophes".

line 47-48. need to be more specific about the types of extreme events being referred to. Suggest rewording this sentence to specify marine heatwaves since they are the focus of Smale et al. 2019 or mentioning additional types of extreme events and with appropriate references.

line 49-50. How often have extreme disturbances led to "species" extinction (as opposed to the extinction of local populations?) in the ocean? My sense is not very often if ever. Consider rewording to avoid this inference.

Line 58. Including ECEs in this sentence is vague and somewhat distracting as this entire paragraph is about MHWs. Also stating that ECE (i.e., extreme climate events) "are some of the extreme climate disturbances" seems redundant and conveys little information. Consider deleting ECE from this sentence and focus on MHWs.

Line 193-198 and 266-270. It's not too surprising that year* depth interaction for nearly all taxonomic groups was significant as normal inter-annual and spatial variability could cause percent cover to vary asynchronously among sites and years. Yet a significant year*depth interaction does not in of itself identify the 2011 MHW as the cause of the significant interaction. Treating "period" (i.e., 2010 to 2011 (heatwave) and 2010 to last survey as in figure 4) as a fixed factor rather than year in the PERMANOVA and using the difference in cover between periods as the response variable should provide better insight into whether the changes in cover reflected changes due to the heatwave vs. more typical asynchronous variation among years and depths that is unrelated to the heatwave.

Line 213-215. Fig 2a does not contain taxon specific information. Fig. 2b shows "coral" and "encrusting" but no "encrusting red algae". Edit text and/or figures so that they consistent with one another.

Lines 215-265. The main and supplemental figures are too coarse for the reader to determine the level of change that occurred. To make it easy on the reader I strongly suggest the authors add the % change in parentheses in the text following the terms "increase", "decrease" and "change".

Figure 4 is a nice addition to the paper, however it may be perceived as a bit misleading as in several instances taxa whose change in cover is near zero are shown as increasing or decreasing. Rather than

simply showing increases and decreases the figure should categorize the changes as either: (1) a significant increase, (2) a significant decrease, or (3) a non-significant change. Only then can the reader evaluate whether the changes are significant. Also I assume that the change in cover is the absolute change rather than the relative change. This should be specified in the figure legend to avoid confusion. What has been deleted from this version is information on the actual cover of the different functional groups. Including this information in the supplement would give the reader much needed understanding of the community structure of the study systems and help them determine whether a small change in cover represents a small change in an abundant taxon or a small change in an uncommon taxon.

Line 229. Check labels on the supplemental figures. Two different figures in the supplement are labeled S8 and two different figures are labeled S9.

Line 235. How large of a decline in seagrasses? Again the authors should use quantitative terms to describe their results in the text rather than force the reader to go to an online supplement for basic yet important information.

Lines 241-242 and 367-368. Need to provide information on what happened to the community in 2017 and how its composition changed.

Line 244. The said decrease in fine branching red algae is not evident in figure S4.

Lines 271-283. Unfortunately, the temperature record is very spotty and contributes little to understanding how temperature contributed to the changes observed in the biota. The different types of temperature data analyzed (e.g., periodic data from CTD casts, AUV deployments and moored sensors) recorded data opportunistically at very different frequencies and time scales (averages were calculated from as little as 0-2 readings per depth for some sites and years) and not enough information is provided to determine the extent to which differences among years reflect differences in the timing and frequency of the different types of measurements, or how data measured at different frequencies and time periods were used to calculate monthly averages. Moreover, the large number of missing values and limited sample size do not instill much confidence in the calculated anomalies, which are relatively small given that 2011 MHW is reported to be the most extreme MHW on record. The fact that the anomalies during the 2011 MHW were greatest at the deeper sites reinforces skepticism (whether justifiable or not) in these data, and suggests that exceedance of a temperature threshold rather than deviation from average conditions might be a more likely cause for the observed changes in the biota following the 2011 MHW. Unfortunately, the temperature data are insufficient to evaluate whether maximum temperatures exceeded the tolerances of the species examined. I understand the author's desire to show temperature records to support their claims, but because they are inadequate, they serve to confuse or raise suspicion about the causes of changes in the biota rather than clarify them. I think the authors would be better served if they simply cited other studies when discussing how temperature changed during the heatwave. If they feel it is necessary to present temperature data from their sites, then they should only present temperature data from Rottneast Is. where the record is complete. If multiple data sources were used to complete the temperature record at this site, then the authors need to provide a more detailed explanation of how the different data sources were integrated in their calculations of average temperature.

Line 311-312. It might be worth noting that deep reef communities only have the potential to be a key driver of shallow reef resilience if the future frequency and intensity of MHW is not so severe to prohibit recovery in shallow water.

Line 370-372. Should clarify that deep populations do not serve as a resilience mechanism for obligate shallow species, but rather for species that occupy a broad depth range. (e.g. 15-40 m).

Line 385-387. Suggest rephrasing since this statement as written can be interpreted to mean that the primary (or only) reason for additional research is because of the high economic value of the lobster fishery.

Response to Referee's comments on Manuscript ID RSPB-2020-0709

We are grateful to the referees for the additional comments and suggestions which have further improved this manuscript. We have addressed all the comments and made the revisions suggested. Specifically, we have explored the change to the PERMANOVA analysis suggested by Referee 3, we have added information about which changes in percent cover of main macroalgae (illustrated in figure 4) were significant in response to the marine heatwave and to the latest survey, and we have removed the temperature data analysis from the main manuscript. We have also addressed all the other comments, which mainly requested clarification of certain aspects of the text and we believe helped made the text clearer for the reader. Below, we provide detailed responses to each comment and their location in the clean manuscript.

Response to Referee 3

No.	Comment / Revision	Author response
1	Line 31. The specific ecological catastrophe referred to in this sentence is unclear. Suggest replacing "this rapid ecological catastrophe" with "such ecological catastrophes".	As suggested we have replaced 'this rapid' with 'such'. The sentence now reads: "To address the response of taxa to such ecological catastrophe, ..." (Line 39).
2	line 47-48. need to be more specific about the types of extreme events being referred to. Suggest rewording this sentence to specify marine heatwaves since they are the focus of Smale et al. 2019 or	We have included examples of extreme climatic events (ECE's) and relevant references. The sentence now reads: "In light of the increase in extreme climatic events (such as atmospheric and marine heatwaves, droughts and wildfires) driven by climate change and their catastrophic consequences in marine environments [15–18]" (Lines 47-48)

Depth moderate loss of marine foundation species after an extreme marine heatwave: Could deep temperate reefs act as refuge?

	mentioning additional types of extreme events and with appropriate references.	
3	line 49-50. How often have extreme disturbances led to “species” extinction (as opposed to the extinction of local populations?) in the ocean? My sense is not very often if ever. Consider rewording to avoid this inference.	As suggested, we have reworded this sentence. The sentence now reads: “they have the potential to prevent the extinction of local populations associated with extreme disturbance events” (Lines 50-51)
4	Line 58. Including ECEs in this sentence is vague and somewhat distracting as this entire paragraph is about MHWs. Also stating that ECE (i.e., extreme climate events) “are some of the extreme climate disturbances” seems redundant and conveys little information. Consider deleting ECE from this sentence and focus on MHWs.	We have removed the mention to ECE’s in this section. The sentence now reads: “Oceanic marine heatwaves (MHWs) are extreme climatic disturbances that are predicted to increase in frequency and intensity due to climate change [8].” (Lines 59-60)
5	Line 193-198 and 266-270. It’s not too surprising that year* depth interaction for nearly all taxonomic groups was significant as normal inter-annual and spatial variability could cause percent cover to vary asynchronously among sites and years. Yet a significant year*depth interaction does not in of itself identify the 2011 MHW as the cause of the significant interaction. Treating “period” (i.e., 2010 to 2011	We thank the referee for this suggestion which we have explored in depth by conducting further PERMANOVA analyses. We used two sets of data to perform univariate PERMANOVA analyses for each benthic class. Both datasets used change in percent cover (difference in mean percent cover between two years) as the response variable (as suggested by the referee) and depth and period as fixed factors. Importantly, change in percent cover was calculated by averaging the percent cover for each grid for each year (~30 images per grid), so we could calculate the difference in mean percent cover of each grid from one year to the next. This resulted in a smaller number of samples compared to the original PERMANOVA analyses that are currently in the manuscript, which used percent cover, not the change in percent cover, as the

Depth moderate loss of marine foundation species after an extreme marine heatwave: Could deep temperate reefs act as refuge?

(heatwave) and 2010 to last survey as in figure 4) as a fixed factor rather than year in the PERMANOVA and using the difference in cover between periods as the response variable should provide better insight into whether the changes in cover reflected changes due to the heatwave vs. more typical asynchronous variation among years and depths that is unrelated to the heatwave.	response variable, and therefore used individual images as replicates. As a result, the new analyses have a much lower level of replication, and on several occasions the same grid was not surveyed each year, which weakened the inference of the analyses. The first data set, tested for differences of change in percent cover between two periods: 2010 to 2011 and 2010 to the last survey, as suggested by the referee (Appendix table A1). The second set, tested for differences of change in percent cover between each consecutive year (2010 to 2011, 2011 to 2012, 2012 to 2013, 2013 to 2014 – for Abrolhos only, 2014 to 2017 – for Abrolhos only, and 2013 to 2017 for Rottneest only)(Appendix table A2). The results from each of these PERMANOVA analyses are in an appendix to this document (Appendix tables A1 and A2). In both PERMANOVA sets, depth often results as a significant factor, and rarely does period, or interaction between depth and period result as significant (Appendix tables A1 and A2). It is important to note, that there are large differences of variance of the response variable across depth, with shallow sites presenting larger variance than deep sites. Most of these differences are significant as shown by PERMDISP analyses also conducted for both data sets (significantly different variances across depth signalled with ‘^’ next to the depth factor, Appendix tables A1 and A2). This lack of homoscedasticity of variances, and the reduced replication resulted in reduced robustness of the PERMANOVA analyses. The large difference in variance across depth can also be observed in the PCO analysis (figure 3) in the manuscript, though this is for the community composition rather than individual classes. Since the PERMANOVA analyses are supplementary information to this study and the inference of the proposed PERMANOVA is largely weakened (as explained above), we have decided to keep the original PERMANOVA analyses that are currently part of the
--	--

Depth moderate loss of marine foundation species after an extreme marine heatwave: Could deep temperate reefs act as refuge?

		manuscript, which provide a worthy description of the dynamics of benthic species in this region. Importantly, the main result of this manuscript is not reflected in these PERMANOVA analyses (hence, it is located in the supplementary material). The main results of this study are illustrated in figures 3 and 4. They show that community composition in deeper sites responded less to the marine heatwave (figure 3) and that important macroalgae species, which were reported to have significant changes in cover after the 2011 marine heatwave in shallow reefs (Wernberg et al., 2013) did not show the same magnitude of change at any depth in offshore reefs. If the editor or referees believe that these PERMANOVAs are informative to this manuscript, we will be happy to include them in the supplementary material.
6	Line 213-215. Fig 2a does not contain taxon specific information. Fig. 2b shows "coral" and "encrusting" but no "encrusting red algae". Edit text and/or figures so that they consistent with one another.	We have eliminated the reference to figure 2a and reference it in a different part of the text, where the community composition of sites at Jurien Bay is compared to the community composition of 25 and 40 m sites at Abrolhos Islands. We have modified the text and specified 'fine branching red algae' to be consistent with figure 2a. (Lines 207-209) Also to clarify, 'coral encrusting' in figure 2b refers to coral with encrusting morphology (other coral morphologies are also referenced in this figure and figures 3a-3f).
7	Lines 215-265. The main and supplemental figures are too coarse for the reader to determine the level of change that occurred. To make it easy on the reader I strongly suggest the authors add the % change in parentheses in the text following the terms "increase", "decrease" and "change".	Thank you for this suggestion. We agree that this will make the text more clear to the readers. We have included approximate percent cover decreases or increases in parenthesis as suggested (Lines 213-253). This section of text now reads: "At Abrolhos islands, there was a trend of greater change across years in community composition at 15 and 25 m, and less at 40 m (figure 3a). The only convergent community composition among years was shown between 2010 and 2017 at the 15 m site at Abrolhos. Following the marine heatwave (2010 to 2011), the 15 m sites of Abrolhos islands changed in community composition with an increase in bleached coral (~4%)

Depth moderate loss of marine foundation species after an extreme marine heatwave: Could deep temperate reefs act as refuge?

		(electronic supplementary material, figure S2) and turf matrix (~20%) (figure 4b, electronic supplementary material, figure S3), and a decrease in fine branching red algae (~11%) (electronic supplementary material, figure S4) and foliose coral (~5%) (electronic supplementary material, figure S5). Minimal change in encrusting red algae cover (~2%) was seen in shallow sites (Figure 4c, electronic supplementary material, figure S6) and decrease in E. radiata (~3%) was observed (figure 4a, electronic supplementary material, figure S7). By 2017, the shallow (15 m) benthic community of Abrolhos Islands appeared to have returned to a state similar to pre-heatwave composition (figure 3a). The 25 m sites showed a response to the heatwave with an increase in Sargassum sp. (~3%), E. radiata (~10%) and seagrass (~5%) (electronic supplementary material, figures S8, S9 and S7 respectively). By 2017, community composition had not returned to pre-heatwave conditions, with reduced turf cover (~15%), increased encrusting red algae (~3%), Sargassum sp. (~3%), and seagrass (~3%) (figure 3a). On the other hand, there was minimal benthic community change at the Abrolhos 40 m sites (figure 3a) after the 2011 MHW and between all the years. The Jurien Bay assemblage also changed across years, with the largest change occurring at the shallower sites (figure 3c) after the marine heatwave. In contrast with the other locations, shallow sites at Jurien bay were more characterised by sand patches and seagrass. Seagrasses at this location showed large declines after the marine heatwave (from ~ 8 % to almost 0% cover) and no signs of recovery at 15 and 25 m sites (electronic supplementary material, figure S9). The 25 and 40m sites at Jurien, also showed change after the marine heatwave, mostly characterised by increases in encrusting algae (~5% increase at both depths) (figure 4c). Communities at 25 m seemed to be affected by a separate event to the 2011 heatwave, since community composition was recovered to pre-heatwave conditions in 2012 and by 2013 it show change towards a more turf driven community (with an increase of ~20%). The community composition at Rottneest Island also responded to the marine heatwave at the shallow sites (15 m), with a reduced response in deeper sites (figure 3e). At the 15 m sites there was a decrease in S. dorycarpa and E. radiata (~5%, from 2010 to 2011 for
--	--	---

Depth moderate loss of marine foundation species after an extreme marine heatwave: Could deep temperate reefs act as refuge?

		both species) (figures 4a and 4d), and an increase in encrusting red algae (~5%) (figure 4c). In contrast to the other two locations, community composition at shallow sites in Rottneest Island had not recovered to pre-heatwave conditions by 2017 (figure 3e). Cover of S. dorycarpa continued to decrease (~10% decrease by 2013) and had not recovered by 2017 (figure 4d, electronic supplementary material, figure S10). Moreover, the analysis also identified changes in community composition in deeper habitats (25 and 40 m) that appeared to be a response to a process separate from the 2011 marine heatwave, as they were observed from 2013 to 2017 (figure 3e). These changes were driven by an increase of ~5% encrusting red algae in the 25 m sites and an increase of ~ 4% in the cover of sponges and S. dorycarpa at 40 m sites (figures 3e, 3f and electronic supplementary material, figures S7, S10 and S11).”
8	Figure 4 is a nice addition to the paper, however it may be perceived as a bit misleading as in several instances taxa whose change in cover is near zero are shown as increasing or decreasing. Rather than simply showing increases and decreases the figure should categorize the changes as either: (1) a significant increase, (2) a significant decrease, or (3) a non-significant change. Only then can the reader evaluate the whether the changes are significant. Also I assume that the change in cover is the absolute change rather than the relative change. This should be specified in the figure legend to avoid confusion.	We also believe Fig. 4 helps to highlight one of the most important findings of this study. We agree that for completeness, it is important to show which of the changes to the percent cover illustrated in this figure are significant. To address this, we conducted a series of one-way analyses of variance (ANOVA), for each location, depth, and species or benthic class illustrated in figure 4. The ANOVAs compared the percent cover between 2010, 2011, and the last survey (which varied with location) of each species or benthic class (in figure 4). In this way, the changes could be compared to the initial percent cover (in 2010), which in some cases was near zero, as pointed by the referee. The results from these analyses are summarized in the supplementary table S2 (the table is also in appendix of this document, Appendix table A3). Details of these analyses have been added to the methods section (Lines 178-186): “These plots were made by calculating the mean percent cover of each species (or benthic class, like turf and encrusting red algae) at each grid, per location, depth and year, and then calculating the absolute change in percent cover from 2010 to 2011 and from 2010 to the last survey (which varied with location, see electronic supplementary material

What has been deleted from this version is information on the actual cover of the different functional groups. Including this information in the supplement would give the reader much needed understanding of the community structure of the study systems and help them determine whether a small change in cover represents a small change in an abundant taxon or a small change in an uncommon taxon.	table S1). Differences in percent cover for E. radiata, turf, encrusting red algae and S. dorycarpa for each location and depth, were analysed by one-way analyses of variance (ANOVA) between 2010, 2011 and the year of last survey, followed by a Tukey-test if differences were significant. When assumptions of normality and homogeneity of variance were violated, a Kruskal-Wallis test was used, and Dunn's post-hoc test." The results from these analyses were also added to the results section (Lines 254-273): "Despite changes in percent cover of macroalgae following the marine heatwave, these were not at the scale of the changes reported for inshore reefs (figure 4) [3]. Decreases of approximately 30% were reported in the cover E. radiata due to the MHW at shallow inshore reefs (figure 4a). We found the largest decrease in E. radiata cover at the deep sites of Rottneest Island to be of ~18% following the 2011 MHW, and at no location or depth where these changes found to be significant (figure 4a). Turf cover increased in around 10% at shallow, inshore reefs after the 2011 heatwave [3], but we only found a comparable increase at the shallow sites of Abrolhos with a significant increase in turf cover of ~ 15% and at the 25 m sites of Jurien with a significant increase of ~8% (figure 4b). Other sites and depths did not show a large increase in turf cover after the 2011 MHW and a large significant decrease in turf cover (~20%) was found in the deep sites of Jurien from 2010 to the last survey in 2017 (figure 4b). The largest decrease in encrusting red algae we observed at the 25 m sites at Abrolhos island, with a reduction of ~ 5%, but this was not significant, while in shallow inshore sites the reductions were of ~ 15 % (figure 4c). Significant increases in encrusting red algae cover were observed in the 15 m and 40 m sites of Abrolhos (~5% at both depths), and at the 40 m sites of Jurien with an increase of ~5% after the heatwave and a total of 10% by the time of the last survey in 2013 (figure 4c). Scytothalia dorycarpa at Rottneest Island showed the largest reduction at shallow sites (~ 5%) and in the last survey it presented a significant decrease (~10 %) compared to pre-heatwave levels, yet these reductions are small compared to the ~ 40% percent cover decrease at inshore reefs (figure 4d)."
---	--

Depth moderate loss of marine foundation species after an extreme marine heatwave: Could deep temperate reefs act as refuge?

		The results from these analyses have also been illustrated in figure 4 with by adding the symbol ‘★’ to the periods where the change in percent cover was significant (the modified figure has also been included in this document as Appendix figure A1). The referee is correct, the change calculated for figure 4 is absolute change, and this information has also been included in the figure legend. The set of figures which made figure 4 in previous version of this manuscript were removed from the main manuscript, but were included in the supplementary material (Supplementary figures S4, S6, S7 and S10), since as pointed by the referee, they constitute important information for the reader to refer to.
9	Line 229. Check labels on the supplemental figures. Two different figures in the supplement are labeled S8 and two different figures are labeled S9.	Thank you for pointing this out. The figure labels in the supplementary material have been fixed.
10	Line 235. How large of a decline in seagrasses? Again the authors should use quantitative terms to describe their results in the text rather than force the reader to go to an online supplement for basic yet important information.	The change in seagrass percent cover has been added to this sentence, which now reads: ‘Seagrasses at this location showed large declines after the marine heatwave (from ~ 8 % to almost 0% cover) and no signs of recovery at 15 and 25 m sites’. (Lines 233-235)
11	Lines 241-242 and 367-368. Need to provide information on what happened to the community in 2017 and how its composition changed.	We have included information on how community composition did not recover or changed in 2017 for each of these sentences. The text now reads: (Lines 227-230): “By 2017, community composition had not returned to pre-heatwave conditions, with reduced turf cover (~15%), increased encrusting red algae (~3%), Sargassum sp. (~3%), and seagrass (~3%) (figure 3a). On the other hand, there was

Depth moderate loss of marine foundation species after an extreme marine heatwave: Could deep temperate reefs act as refuge?

		minimal benthic community change at the Abrolhos 40 m sites (figure 3a) after the 2011 MHW and between all the years.” (Lines 362-365): “Yet, our understanding of the processes driving the community dynamics of deeper reefs is still in its infancy, as indicated by the large change we observed in community composition at 40 m sites of Rottneest Island in 2017, mainly driven by an increase in sponges which we were unable to associate with an environmental change or disturbance.”
12	Line 244. The said decrease in fine branching red algae is not evident in figure S4.	Thank you for pointing this mistake. There was no evident change in fine branching, instead we have made reference to the increases in E. radiata , turf and encrusting red algae shown in figures 4a-4c. (Lines 245-247)
13	Lines 271-283. Unfortunately, the temperature record is very spotty and contributes little to understanding how temperature contributed to the changes observed in the biota. The different types of temperature data analyzed (e.g., periodic data from CTD casts, AUV deployments and moored sensors) recorded data opportunistically at very different frequencies and time scales (averages were calculated from as little as 0-2 readings per depth for some sites and years) and not enough information is provided to determine the extent to which differences among years reflect differences in the timing and frequency	We thank the referee for this observation and suggestion. It is true that the temperature data is very patchy and it has a large number of missing values, and at times even just one value per depth, per season. Despite being the best data set available, unfortunately, it does not provide robust evidence of the temperature conditions at depth at all years. As such, we believe it is best to remove this figure from the main manuscript as suggested by the referee. However, we have decided to include the figure and summary tables of data used to make the figure in the supplementary material (Supplementary figure S14, and Supplementary tables S4, S5, and S6). This is to reassure readers that we have collected the best records of temperature available for the region, and attempted to use them for analysis. Also, they describe similar patterns to other studies of the 2011 marine heatwave in WA, where there the temperature anomaly signals were detected down to 50 m of depths (Pearce and Feng, 2013; Benthuisen et al 2014).

Depth moderate loss of marine foundation species after an extreme marine heatwave: Could deep temperate reefs act as refuge?

of the different types of measurements, or how data measured at different frequencies and time periods were used to calculate monthly averages. Moreover, the large number of missing values and limited sample size do not instill much confidence in the calculated anomalies, which are relatively small given that 2011 MHW is reported to be the most extreme MHW on record. The fact that the anomalies during the 2011 MHW were greatest at the deeper sites reinforces skepticism (whether justifiable or not) in these data, and suggests that exceedance of a temperature threshold rather than deviation from average conditions might be a more likely cause for the observed changes in the biota following the 2011 MHW. Unfortunately, the temperature data are insufficient to evaluate whether maximum temperatures exceeded the tolerances of the species examined. I understand the author's desire to show temperature records to support their claims, but because they are inadequate, they serve to confuse or raise suspicion about the causes of changes in the biota rather than clarify them. I think the authors would be better	The discussion section where we describe the results from the temperature data, has been changed, now we refer only to other studies that have revised the effects of the 2011 marine heatwave across depth. This section now reads (Lines 326-336): “While temperature anomalies associated with the 2010-2011 marine heatwave have been identified down to ~50-60 m of depth [33,55], we did not detect signs of catastrophic alteration in community composition as documented in shallower habitats (less than 15 m), as far south as Rottneest Island. We gathered sporadic in situ temperature recordings near our study sites over a 20 year period, which also show temperature anomalies at 40 m depths during the 2011 MHW (electronic supplementary material figure S14), however these data lacked of enough replication over time to be used for further analyses. Benthic populations living in deeper reefs may be acclimated to frequent thermal variation due to the effect of the Leeuwin Current which transports warm water from the tropics along the continental shelf of Western Australia [56] and consequently may have greater influence in deep offshore habitats than in shallow and inshore ones [57].”
---	--

Depth moderate loss of marine foundation species after an extreme marine heatwave: Could deep temperate reefs act as refuge?

	served if they simply cited other studies when discussing how temperature changed during the heatwave. If they feel it is necessary to presented temperature data from their sites, then they should only present temperature data from Rottneest Is. where the record is complete. If multiple data sources were used to complete the temperature record at this site, then the authors need to provide a more detailed explanation of how the different data sources were integrated in their calculations of average temperature.	
14	Line 311-312. It might be worth noting that deep reef communities only have the potential to be a key driver of shallow reef resilience if the future frequency and intensity of MHW is not so severe to prohibit recovery in shallow water.	We have addressed this comment and the text now reads (Lines 305-308): “Since MHWs are predicted to become more frequent and intense in the future [8,9], deep reef communities may be a key driver of shallow-reef resilience, inasmuch as the frequency and magnitude of future marine heatwaves allows for the recovery of shallow communities.”
15	Line 370-372. Should clarify that deep populations do not serve as a resilience mechanism for obligate shallow species, but rather for species that occupy a broad depth range. (e.g. 15-40 m).	We have addressed this comment and the text now reads (Lines 367-369): “The response of deep communities to future extreme events needs to be evaluated to confirm their role as a resilience mechanism for depth generalist species living in shallow reefs.”
16	Line 385-387. Suggest rephrasing since this statement as written can be interpreted to mean that the	We have addressed this comment and the text now reads (Lines 383-385):

Depth moderate loss of marine foundation species after an extreme marine heatwave: Could deep temperate reefs act as refuge?

	primary (or only) reason for additional research is because of the high economic value of the lobster fishery.	“Further research into deeper communities is required to fully understand their potential to act as refuges for shallow benthic foundation species and the ecosystem services they provide.”
--	--	--

Appendix

Appendix table A1. Univariate PERMANOVA to test for change in percent cover differences of main benthic categories between depths and periods (2010 to 2011, and 2010 to last survey) at each location. Change in percent cover calculated as the absolute difference in mean percent cover of each grid from one year to the next (according to period). PERMANOVA derived from square root transformation and Euclidian distance similarity matrices. Significant PERMDISP for depth factor indicated as ‘^’ next to depth effect.

Benthic category	Location	Effects	Numerator df	Denominator df	Pseudo-F	p	Unique permutations
Coral	Abrolhos	Depth	2	36	5.4467	0.007	9955
		Period	1	36	7.73 x 10 ⁻⁰²	0.7844	9846
		Depth * Period	2	36	8.65 x 10 ⁻⁰³	0.9906	9950
Bleached Coral	Abrolhos	Depth [^]	2	36	4.4598	0.0111	9950
		Period	1	36	3.42	0.0646	9832

Depth moderate loss of marine foundation species after an extreme marine heatwave: Could deep temperate reefs act as refuge?

		Depth *	2	36	2.46	0.0824	9929
Coral - Staghorn	Abrolhos	Depth^	2	36	10.85	0.0004	9848
		Period	1	36	1.53×10^{-01}	0.6989	9848
		Depth *	2	36	1.46×10^{-01}	0.8678	9940
Coral - Foliose	Abrolhos	Depth^	2	36	5.8823	0.0055	9952
		Period	1	36	6.97×10^{-02}	0.7888	9845
		Depth *	2	36	5.49×10^{-02}	0.9514	9948
Coral - Tabulate	Abrolhos	Depth^	2	36	22.579	0.0001	9953
		Period	1	36	8.81×10^{-02}	0.7654	9816
		Depth *	2	36	2.09×10^{-02}	0.9793	9961
Macroalgae	Abrolhos	Depth	2	36	22.579	0.0001	9953
		Period	1	36	8.81×10^{-02}	0.7654	9816
		Depth *	2	36	2.09×10^{-02}	0.9793	9961
	Rottneest	Depth	2	30	2.5322	0.0921	9938
		Period	1	30	1.75	0.2041	9820

Depth moderate loss of marine foundation species after an extreme marine heatwave: Could deep temperate reefs act as refuge?

		Depth *	2	30	6.33×10^{-02}	0.9392	9958
	Jurien	Depth	2	42	0.13428	0.8652	9955
		Period	1	42	9.22×10^{-02}	0.7551	9843
		Depth *	2	42	4.72	0.014	9957
		Period	1	42			
Macroalgae - Ecklonia	Abrolhos	Depth^	2	36	3.706	0.0353	9950
		Period	1	36	1.35×10^{-05}	0.9974	9842
		Depth *	2	36	1.74×10^{-01}	0.8461	9940
		Period	1	36			
	Rottnest	Depth	2	30	0.67181	0.5152	9962
		Period	1	30	1.13	0.2926	9823
		Depth *	2	30	9.93×10^{-02}	0.9041	9944
		Period	1	30			
	Jurien	Depth^	2	42	2.6186	0.0791	9954
		Period	1	42	4.29×10^{-02}	0.8369	9841
		Depth *	2	42	3.59×10^{-01}	0.7126	9953
		Period	1	42			
Macroalgae - Sargassum	Abrolhos	Depth^	2	36	2.8265	0.0751	9946
		Period	1	36	2.12×10^{-01}	0.6506	9837

Depth moderate loss of marine foundation species after an extreme marine heatwave: Could deep temperate reefs act as refuge?

		Depth *	2	36	6.77×10^{-02}	0.9354	9957
		Period					
	Rottnest	Depth^	2	30	2.6148	0.0901	9944
		Period	1	30	1.03×10^{-01}	0.757	9855
		Depth *	2	30	2.89×10^{-02}	0.9743	9938
		Period					
Macroalgae - Turf	Abrolhos	Depth	2	36	8.1555	0.0014	9954
		Period	1	36	9.49×10^{-01}	0.3259	9856
		Depth *	2	36	2.81×10^{-01}	0.7533	9952
		Period					
	Rottnest	Depth	2	30	0.33215	0.7105	9965
		Period	1	30	2.46	0.1256	9843
		Depth *	2	30	1.14	0.3347	9955
		Period					
	Jurien	Depth^	2	42	2.7529	0.0805	9955
		Period	1	42	1.54×10^{-01}	0.6926	9845
		Depth *	2	42	9.82×10^{-02}	0.9078	9950
		Period					
Macroalgae - Fine branching red	Abrolhos	Depth	2	36	5.6297	0.0069	9952
		Period	1	36	2.43×10^{-02}	0.8725	9830

Depth moderate loss of marine foundation species after an extreme marine heatwave: Could deep temperate reefs act as refuge?

		Depth * Period	2	36	5.65×10^{-01}	0.5764	9953
	Rottneest	Depth^	2	30	2.5487	0.0925	9940
		Period	1	30	6.33	0.0152	9841
		Depth * Period	2	30	7.59×10^{-01}	0.4777	9955
	Jurien	Depth	2	42	0.36225	0.6911	9941
		Period	1	42	3.54	0.0664	9853
		Depth * Period	2	42	2.97	0.0605	9954
Macroalgae – Encrusting red	Abrolhos	Depth^	2	36	15.892	0.0002	9948
		Period	1	36	3.24	0.0855	9827
		Depth * Period	2	36	7.35×10^{-01}	0.4817	9955
	Rottneest	Depth	2	30	0.46433	0.634	9956
		Period	1	30	3.77×10^{-01}	0.5484	9845
		Depth * Period	2	30	6.90×10^{-01}	0.5051	9972
	Jurien	Depth^	2	42	43.227	0.0001	9941
		Period	1	42	2.97×10^{-05}	0.9957	9819

Depth moderate loss of marine foundation species after an extreme marine heatwave: Could deep temperate reefs act as refuge?

		Depth *	2	42	3.72	0.0334	9958
		Period					
Macroalgae - Scytothalia	Rottneest	Depth	2	30	3.3357	0.0499	9944
		Period	1	30	7.28×10^{-01}	0.3997	9855
		Depth *	2	30	1.71	0.2006	9946
		Period					
Macroalgae - Canopy	Abrolhos	Depth^	2	36	4.3251	0.0209	9951
		Period	1	36	3.77×10^{-02}	0.8437	9826
		Depth *	2	36	2.07×10^{-01}	0.8163	9956
		Period					
	Rottneest	Depth	2	30	0.23827	0.7906	9954
		Period	1	30	3.93	0.054	9823
		Depth *	2	30	1.79×10^{-02}	0.9834	9942
		Period					
	Jurien	Depth^	2	42	2.7443	0.0688	9954
		Period	1	42	1.28×10^{-02}	0.9128	9819
		Depth *	2	42	1.10	0.3518	9937
		Period					
Seagrass	Abrolhos	Depth	2	36	1.7413	0.1753	9943
		Period	1	36	1.28×10^{-01}	0.7303	9846

Depth moderate loss of marine foundation species after an extreme marine heatwave: Could deep temperate reefs act as refuge?

		Depth * Period	2	36	3.51×10^{-02}	0.9705	9955
	Jurien	Depth	2	42	0.9588	0.4024	9968
		Period	1	42	1.28×10^{-01}	0.7266	9828
		Depth * Period	2	42	1.87×10^{-01}	0.831	9952
Sponges	Abrolhos	Depth^	2	36	7.151	0.0034	9940
		Period	1	36	1.24×10^{-02}	0.9135	9823
		Depth * Period	2	36	5.50×10^{-03}	0.9939	9952
	Rottnest	Depth	2	30	8.6182	0.0013	9944
		Period	1	30	2.38×10^{-01}	0.6293	9822
		Depth * Period	2	30	1.86×10^{-01}	0.8252	9952
	Jurien	Depth^	2	42	18.644	0.0001	9938
		Period	1	42	1.25	0.2793	9827
		Depth * Period	2	42	2.77×10^{-01}	0.7711	9943
Sand	Abrolhos	Depth^	2	36	27.534	0.0001	9945
		Period	1	36	5.67	0.0233	9834
		Depth * Period	2	36	3.36	0.0492	9953
	Rottnest	Depth	2	30	8.7231	0.0014	9948

Depth moderate loss of marine foundation species after an extreme marine heatwave: Could deep temperate reefs act as refuge?

		Period	1	30	4.68×10^{-01}	0.5013	9821
		Depth * Period	2	30	2.22×10^{-01}	0.8023	9950
	Jurien	Depth^	2	42	4.2422	0.0212	9951
		Period	1	42	1.61×10^{-03}	0.9647	9824
		Depth * Period	2	42	3.06×10^{-01}	0.7429	9960

Appendix table A2. Univariate PERMANOVA to test for change in percent cover differences of main benthic categories between depths and periods (2010 to 2011, 2011 to 2012, 2012 to 2013, 2013 to 2014 – for Abrolhos only, 2014 to 2017 – for Abrolhos only, and 2013 to 2017 for Rottneest only). at each location. Change in percent cover calculated as the absolute difference in mean percent cover of each grid from one year to the next (according to period). PERMANOVA derived from square root transformation and Euclidian distance similarity matrices. Significant PERMDISP for depth factor indicated as ‘^’ next to depth effect.

Benthic category	Location	Effects	Numerator df	Denominator df	Pseudo-F	p	Unique permutations
Coral	Abrolhos	Depth^	2	90	32.52	0.0001	9948
		Period	4	90	0.15551	0.961	9953
		Depth * Period	8	90	0.41535	0.9075	9916
Bleached Coral	Abrolhos	Depth^	2	90	10.908	0.0001	9959

Depth moderate loss of marine foundation species after an extreme marine heatwave: Could deep temperate reefs act as refuge?

		Period	4	90	1.3224	0.2597	9960
		Depth * Period	8	90	0.87623	0.5415	9934
Coral - Staghorn	Abrolhos	Depth^	2	90	26.483	0.0001	9942
		Period	4	90	0.92646	0.4489	9953
		Depth * Period	8	90	0.88234	0.5443	9932
Coral - Foliose	Abrolhos	Depth^	2	90	12.243	0.0001	9947
		Period	4	90	1.0825	0.3622	9946
		Depth * Period	8	90	0.46783	0.879	9942
Coral - Tabulate	Abrolhos	Depth^	2	90	54.92	0.0001	9951
		Period	4	90	0.97126	0.4203	9948
		Depth * Period	8	90	0.83286	0.5853	9952
Macroalgae	Abrolhos	Depth	2	90	6.0286	0.0036	9952
		Period	4	90	4.8148	0.0014	9942
		Depth * Period	8	90	4.327	0.0003	9950
	Rottnest	Depth	2	60	2.1479	0.1268	9951

Depth moderate loss of marine foundation species after an extreme marine heatwave: Could deep temperate reefs act as refuge?

		Period	3	60	1.4413	0.2405	9954
		Depth * Period	6	60	0.50117	0.8104	9944
	Jurien	Depth^	2	63	2.3847	0.1014	9953
		Period	2	63	3.2247	0.0523	9946
		Depth * Period	4	63	0.60637	0.653	9946
Macroalgae - Ecklonia	Abrolhos	Depth^	2	90	6.3725	0.0029	9956
		Period	4	90	1.4291	0.2285	9954
		Depth * Period	8	90	0.54378	0.8249	9937
	Rottnest	Depth	2	60	0.20119	0.8213	9951
		Period	3	60	2.7931	0.0444	9962
		Depth * Period	6	60	0.53994	0.7792	9940
	Jurien	Depth^	2	63	8.1613	0.0009	9954
		Period	2	63	0.26323	0.7651	9964
		Depth * Period	4	63	0.44708	0.7739	9942
Macroalgae - Sargassum	Abrolhos	Depth^	2	90	17.057	0.0001	9953
		Period	4	90	1.4162	0.2338	9951

Depth moderate loss of marine foundation species after an extreme marine heatwave: Could deep temperate reefs act as refuge?

		Depth * Period	8	90	1.1416	0.3482	9927
	Rottne	Depth^	2	60	12.205	0.0002	9952
		Period	3	60	1.6379	0.1912	9947
		Depth * Period	6	60	1.1156	0.3743	9951
Macroalgae - Turf	Abrolhos	Depth^	2	90	6.0166	0.0035	9959
		Period	4	90	3.7247	0.0082	9953
		Depth * Period	8	90	1.0396	0.4132	9933
	Rottne	Depth^	2	60	1.1988	0.313	9955
		Period	3	60	1.344	0.2639	9958
		Depth * Period	6	60	1.3702	0.2389	9927
	Jurien	Depth^	2	63	2.5456	0.0877	9948
		Period	2	63	0.28609	0.748	9946
		Depth * Period	4	63	0.78197	0.5433	9951
Macroalgae - Fine branching red	Abrolhos	Depth^	2	90	18.885	0.0001	9956
		Period	4	90	0.69939	0.5913	9945

Depth moderate loss of marine foundation species after an extreme marine heatwave: Could deep temperate reefs act as refuge?

		Depth * Period	8	90	1.0286	0.4225	9940
	Rottnest	Depth^	2	60	6.8011	0.0029	9963
		Period	3	60	1.6092	0.192	9948
		Depth * Period	6	60	0.59791	0.7387	9941
	Jurien	Depth^	2	63	4.4594	0.0145	9948
		Period	2	63	2.5669	0.0894	9944
		Depth * Period	4	63	1.8596	0.1293	9953
Macroalgae – Encrusting red	Abrolhos	Depth^	2	90	4.6036	0.011	9946
		Period	4	90	0.45039	0.7757	9964
		Depth * Period	8	90	1.6833	0.1131	9947
	Rottnest	Depth^	2	60	0.7318	0.4837	9945
		Period	3	60	0.68743	0.5622	9958
		Depth * Period	6	60	0.71792	0.6305	9941
	Jurien	Depth^	2	63	33.212	0.0001	9953
		Period	2	63	1.4568	0.2326	9954

Depth moderate loss of marine foundation species after an extreme marine heatwave: Could deep temperate reefs act as refuge?

		Depth * Period	4	63	0.59639	0.6659	9966
Macroalgae - Scytothalia	Rottneest	Depth	2	60	0.42527	0.6505	9954
		Period	3	60	1.4158	0.2492	9949
		Depth * Period	6	60	0.26251	0.9536	9946
Macroalgae - Canopy	Abrolhos	Depth^	2	90	9.4769	0.0004	9939
		Period	4	90	1.6273	0.1765	9958
		Depth * Period	8	90	1.0759	0.3868	9952
	Rottneest	Depth	2	60	0.17505	0.8462	9942
		Period	3	60	2.1322	0.1033	9951
		Depth * Period	6	60	0.69079	0.6612	9951
	Jurien	Depth^	2	63	5.667	0.0043	9948
		Period	2	63	0.93859	0.4041	9954
		Depth * Period	4	63	0.56146	0.6969	9937
Seagrass	Abrolhos	Depth^	2	90	3.3085	0.0326	9958
		Period	4	90	1.1146	0.3555	9944

Depth moderate loss of marine foundation species after an extreme marine heatwave: Could deep temperate reefs act as refuge?

		Depth * Period	8	90	0.37406	0.9418	9938
	Jurien	Depth	2	63	1.2876	0.2935	9944
		Period	2	63	3.2058	0.039	9947
		Depth * Period	4	63	0.16061	0.9616	9957
Sponges	Abrolhos	Depth^	2	90	7.169	0.0015	9962
		Period	4	90	4.1949	0.0041	9954
		Depth * Period	8	90	0.98949	0.4543	9947
	Rottnest	Depth^	2	60	11.013	0.0002	9953
		Period	3	60	1.1388	0.3396	9954
		Depth * Period	6	60	0.34292	0.9089	9934
	Jurien	Depth^	2	63	33.611	0.0001	9934
		Period	2	63	1.0155	0.3781	9958
		Depth * Period	4	63	0.41114	0.8059	9953
Sand	Abrolhos	Depth^	2	90	26.156	0.0001	9946
		Period	4	90	8.4423	0.0001	9947
		Depth * Period	8	90	2.7025	0.0112	9939
	Rottnest	Depth^	2	60	11.051	0.0004	9950

Depth moderate loss of marine foundation species after an extreme marine heatwave: Could deep temperate reefs act as refuge?

		Period	3	60	1.0608	0.3686	9955
		Depth * Period	6	60	0.14673	0.9879	9933
	Jurien	Depth^	2	63	1.4982	0.2244	9958
		Period	2	63	4.0165	0.0214	9951
		Depth * Period	4	63	3.8116	0.0089	9951

Appendix table S3. Summary statistics of one-way ANOVA for variables of *A. circularis* measured in treatment plots, before and after eelgrass trimming. Values in bold show significant differences between treatment plots. Values marked with * denote non-parametric tests (Kruskal-wallis chi-squared values).

Class	Location	Depth	df	F-value	p-value	Post-hoc
Ecklonia	Abrolhos	15 m	2	0.29732 *	0.8619	
		25 m	2	0.725	0.502	
		40 m	2	2.343	0.158	
	Jurien	15 m	2	0.45226 *	0.7976	
		25 m	2	0.023	0.978	
		40 m	2	0.0042 *	0.9979	
Rottnest	15 m	2	1.131	0.351		
	25 m	2	0.162	0.852		
	40 m	2	0.581	0.575		
Turf	Abrolhos	15 m	2	7.637	<0.01	2010-2011

Depth moderate loss of marine foundation species after an extreme marine heatwave: Could deep temperate reefs act as refuge?

						2010-2014
						2011-2014
		25 m	2	1.952	0.179	
		40 m	2	0.265	0.774	
	Jurien	15 m	2	2.439	0.149	
		25 m	2	28.17	<0.001	2010-2011 2010-2013 2011-2013
		40 m	2	10.017 *	<0.01	2010-2011 2010-2013 2011-2013
	Rottnest	15 m	2	0.792	0.472	
		25 m	2	1.334	0.295	
		40 m	2	0.38	0.692	
Encrusting Red	Abrolhos	15 m	2	10.675 *	<0.01	2010-2011 2010-2014 2011-2014
		25 m	2	1.128	0.351	
		40 m	2	7.799	<0.05	2010-2012 2010-2014 2012-2014
	Jurien	15 m	2	4.6012 *	0.1002	
		25 m	2	1.996	0.186	
		40 m	2	11.26	<0.01	2010-2011

Depth moderate loss of marine foundation species after an extreme marine heatwave: Could deep temperate reefs act as refuge?

						2010-2013
						2011-2013
	Rottne	15 m	2	1.583	0.24	
		25 m	2	0.884	0.435	
		40 m	2	0.47	0.636	
Scytothalia	Rottne	15 m	2	4.262	<0.05	2010-2011
						2010-2017
						2011-2017
		25 m	2	0.241	0.789	
		40 m	2	0.279	0.761	

Depth moderate loss of marine foundation species after an extreme marine heatwave: Could deep temperate reefs act as refuge?

Appendix figure A1. Absolute change in mean percent cover (± SE) from 2010 to 2011 (heatwave) and from 2010 to latest survey for *E. radiata* (a), turf matrix (b), encrusting red algae(c), and *S. dorycarpa* (d) at each location (Houtman Abrolhos Islands, Jurien Bay, Rottnest Island) and depth (15, 25 and 40 m). Colours describe and increase (blue) or decrease (red) in percent cover in relation to 2010. Significant changes in percent cover are marked with '*'. The grey box indicates to the level of change reported for inshore reefs in response to the marine heatwave [3]. The estimates of percent cover are means of 2-6 grids (~30 images per grid) within each location and depth per year.

● Decrease ● Increase

References

- Benthuisen, J., M. Feng, and L. Zhong. 2014. Spatial patterns of warming off Western Australia during the 2011 Ningaloo Niño: Quantifying impacts of remote and local forcing. *Cont. Shelf Res.* **91**: 232–246. doi:10.1016/j.csr.2014.09.014
- Pearce, A. F., and M. Feng. 2013. The rise and fall of the “marine heat wave” off Western Australia during the summer of 2010/2011. *J. Mar. Syst.* **111–112**: 139–156. doi:10.1016/j.jmarsys.2012.10.009
- Wernberg, T., D. A. Smale, F. Tuya, M. S. Thomsen, T. J. Langlois, T. de Bettignies, S. Bennett, and C. S. Rousseaux. 2013. An extreme climatic event alters marine ecosystem structure in a global biodiversity hotspot. *Nat. Clim. Chang.* **3**: 78–82. doi:10.1038/nclimate1627